# Stop Wasting Your Tokens: Towards Efficient Runtime Multi-Agent Systems

**Fulin Lin**[1,*]  **Shaowen Chen**[1]  **Ruishan Fang**[2,1]  **Hongwei Wang**[1,3,†]  **Tao Lin**[2,†]

[1]Zhejiang University    [2]Westlake University
[3]State Key Laboratory of CAD&CG, Zhejiang University

{fulin1.24, hongweiwang}@intl.zju.edu.cn   swenchen@zju.edu.cn
{fangruishan, lintao}@westlake.edu.cn

## Abstract

While Multi-Agent Systems (MAS) excel at complex tasks, their growing autonomy with operational complexity often leads to critical inefficiencies, such as excessive token consumption and failures arising from misinformation. Existing methods primarily focus on post-hoc failure attribution, lacking proactive, real-time interventions to enhance robustness and efficiency. To this end, we introduce SUPERVISORAGENT, a lightweight and modular framework for runtime, adaptive supervision that operates without altering the base agent's architecture. Triggered by an LLM-free adaptive filter, SUPERVISORAGENT intervenes at critical junctures to proactively correct errors, guide inefficient behaviors, and purify observations. On the challenging GAIA benchmark, SUPERVISORAGENT reduces the token consumption of the Smolagent framework by an average of 29.68% without compromising its success rate. Extensive experiments across five additional benchmarks (math reasoning, code generation, and question answering) and various SoTA foundation models validate the broad applicability and robustness of our approach.

## 1 Introduction

The advent of powerful Large Language Models (LLMs) has catalyzed significant advancements in Multi-Agent Systems (MAS) (Liu et al., 2025a; Gao et al., 2025), enabling them to achieve remarkable performance across diverse and challenging domains such as mathematical reasoning (Shang et al., 2025), code generation (Lu et al., 2025), and complex question answering (Luo et al., 2025). This progress has spurred research into sophisticated agent architectures, including self-evolving systems that learn from feedback and experience (Shi et al., 2025b; Liu et al., 2025b), and dynamic topologies that adapt to task complexity (Li et al., 2025a;b). However, a critical paradox has emerged: as these systems grow more capable and complex, they often become less robust and economically viable (Wu et al., 2025a; Huang et al., 2025). Systemic inefficiencies incur prohibitive computational costs, while intricate interactions introduce vectors for unpredictable failures (Zhang et al., 2025e).

This lack of robustness stems from the operational complexity of modern MAS, which introduces a significant reliability challenge (Tian et al., 2025). The long chain of interactions inherent in these systems creates fertile ground for **error propagation** (Dong et al., 2025; Shen et al., 2025). For instance, a single piece of misinformation generated by an agent, a common risk with today's powerful yet occasionally hallucinatory foundation models (Kalai et al., 2025; Farquhar et al., 2024), can be committed to memory and subsequently poison the reasoning of all downstream agents (as explained in Figure 1a). These vulnerabilities mean that even a state-of-the-art MAS can fail on tasks well within its theoretical capabilities, simply due to a lack of operational robustness (Chen et al., 2024).

Furthermore, the issue of **economic inefficiency** is a major barrier to the real-world deployment of MAS (Wang et al., 2025a). We identify two primary sources of this inefficiency. First, agents often struggle with long observations, such as verbose web pages or tool outputs, which flood their context windows. This not only inflates token costs but can also obscure critical information, causing the agent to lose focus and derail its task execution (Hosseini et al., 2025). Second, agents may adopt sub-optimal strategies, entering into repetitive action loops or choosing unnecessarily complex paths to a solution (Cemri et al., 2025), further wasting computational resources (see Figure 1a).

---

*Work was done during Fulin's visit to Westlake University.

†Corresponding authors.

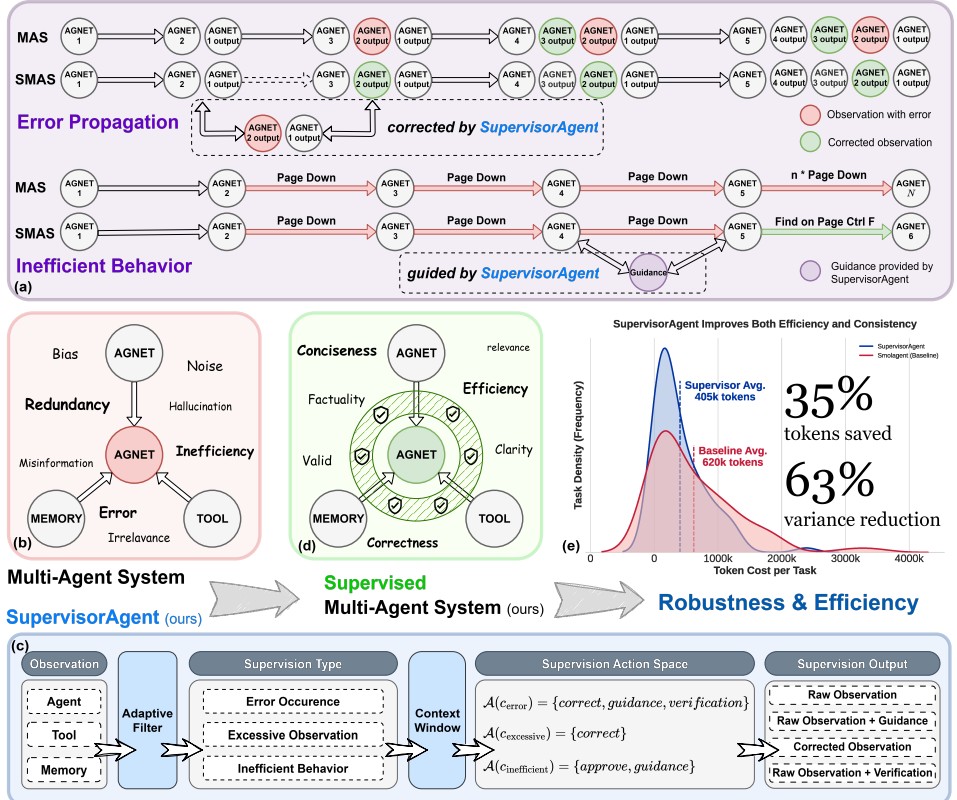

Figure 1: **The SUPERVISORAGENT Framework: Concept and Impact. (a)** Illustrative examples of common failure modes in MAS, including **error propagation** and **inefficient loops**, and the corresponding intervention by our SUPERVISORAGENT. **(b)** An overview of a conventional MAS, highlighting the high-risk interaction loci (agent-agent, agent-tool, agent-memory) where such failures occur. **(c)** The core workflow of our SUPERVISOR-AGENT, which monitors these interactions to provide real-time intervention. **(d)** The resulting Supervised MAS (SMAS), which integrates the SUPERVISORAGENT to enhance robustness and efficiency. **(e)** Performance on GAIA (Level 2), where SMAS (blue) reduces token cost by 35% and variance by 63% versus the baseline (red).

To address these intertwined challenges, we propose SUPERVISORAGENT, a lightweight and modular framework that enhances the **robustness** and **efficiency** of Multi-Agent Systems (MAS) through real-time supervision (see Figure 1c). Incorporating an adaptive filter, SUPERVISORAGENT enables proactive process control, exemplified by its GAIA Level 2 performance in Figure 1e. It adaptively intervenes at critical junctures to mitigate key operational risks: it conducts proactive error diagnosis, provides pragmatic guidance for inefficient behaviors, and performs adaptive observation purification to reduce contextual noise from long observations.

**In summary, our main contributions are:**

1. We propose and implement **SUPERVISORAGENT**, a novel, lightweight, and non-intrusive meta-agent framework for real-time MAS supervision. It improves agent robustness and efficiency through proactive error correction, inefficiency guidance, and adaptive observation purification, without altering the base agents' architecture.

2. We conduct extensive experiments on the challenging **GAIA** benchmark and demonstrate a significant **Pareto improvement**. When applied to the Smolagent framework (Roucher et al., 2025), SUPERVISORAGENT reduces token consumption by an average of **29.68%** while maintaining competitive task success rates.

3. We validate the **general applicability** of our approach across five additional benchmarks spanning mathematical reasoning, code generation, and question answering. Our method consistently delivers substantial efficiency gains, highlighted by a **23.74%** token reduction on HumanEval alongside an accuracy improvement. The framework's effectiveness is further confirmed across various foundation models, including the GPT-4.1, Gemini-2.5-pro, and Qwen3 series.

## 2 RELATED WORK

**The increasing complexity of Multi-Agent Systems (MAS).** Recent advancements in Large Language Models have spurred the development of increasingly sophisticated Multi-Agent Systems (MAS) capable of tackling complex, multi-step tasks (Tran et al., 2025; He et al., 2025). Frameworks like Tongyi DeepResearch (Team, 2025c), AgentOrchestra (Zhang et al., 2025f), and Aime (Shi et al., 2025a) exemplify this trend, introducing complex features such as hierarchical structures (Zhu et al., 2025; Cheng et al., 2025), dynamic agent management (Wu et al., 2025b; Zhang et al., 2025g), and end-to-end training (Li et al., 2025b; Ye et al., 2025). However, this escalating architectural complexity invariably introduces significant challenges in maintaining operational robustness and computational efficiency, which we address in this work.

**Failure attribution and robustness.** A significant body of work has emerged to address the challenge of MAS robustness, primarily focusing on post-hoc *failure attribution* (Zhang et al., 2025e). Systems like Aegis (Song et al., 2025) and SHIELDA (Zhou et al., 2025) propose taxonomies for failure analysis, while AgenTracer (Zhang et al., 2025b) and A2P (West et al., 2025) introduce methods to better trace the root causes of task failures. While valuable, these methods are fundamentally reactive, analyzing failures after they have occurred. In contrast, our SUPERVISORAGENT is designed for *proactive, real-time intervention*, aiming to detect and mitigate high-risk steps *before* they lead to systemic failure.

**Efficient Multi-Agent Systems.** Another stream of research targets the **efficiency** of MAS, a critical factor largely driven by token consumption. Most approaches focus on *design-time optimization*. Some prune the system's architecture by eliminating agents with AgentDropout (Wang et al., 2025b) or communication links with SafeSieve (Zhang et al., 2025d). Others generatively construct efficient prompts (Han et al., 2025) or agent topologies from the outset, as seen in MetaAgent (Zhang et al., 2025h), MaAS (Zhang et al., 2025a), and HiVA (Tang et al., 2025). A second direction, *context compression*, aims to reduce token count by summarizing or distilling observations (Chen et al., 2025; Mou et al., 2025). Our work is orthogonal to these methods. Instead of focusing on static design or message content, we introduce **runtime process control**. SUPERVISORAGENT addresses dynamic inefficiencies *during* execution, a complementary approach that can enhance existing systems.

## 3 PRELIMINARY

In this section, we first establish a formalism for our proposed Supervised Multi-Agent System (SMAS). We then detail the core components of our framework: the SUPERVISORAGENT's action space and the contextual information it leverages for decision-making.

### 3.1 A FORMALISM FOR SUPERVISED MULTI-AGENT SYSTEMS

Our work is predicated on the idea that the complex, often chaotic, interactions within a Multi-Agent System (MAS; see Figure 1b) can be actively managed to improve both robustness and efficiency. To formalize this, we introduce the concept of a Supervised Multi-Agent System (SMAS; see Figure 1d).

> **Definition 1 (Supervised Multi-Agent System (SMAS)) .** *A SMAS is a Multi-Agent System augmented with a meta-level control agent, henceforth referred to as the **Supervisor**. The Supervisor's objective is to monitor agent interactions in real-time, proactively detecting and mitigating operational risks without altering the core logic of the agents it oversees. In this work, we implement this conceptual Supervisor as a concrete agent named **SUPERVISORAGENT**.*

The fundamental unit of supervision is the **interaction**, which occurs when an agent engages with other system components. We categorize interactions into three primary types:

1. **Agent-Agent Interactions:** Communication or delegation between agents. In architectures like ReAct (Yao et al., 2023), where an agent's output becomes another's input, this channel is highly susceptible to the propagation of hallucinated or erroneous information (Shen et al., 2025);
2. **Agent-Tool Interactions:** The invocation of external tools or APIs. This interaction is a primary source of external information, but it is also fraught with risks, including factually incorrect, irrelevant, or outdated data that can corrupt the agent's context (Qian et al., 2025);

3. **Agent-Memory Interactions:** The retrieval of information from short- or long-term memory stores. While crucial for self-evolving systems, memory introduces the hazard of acting upon stale or flawed information from past experiences (Xiong et al., 2025).

## 3.2 THE SUPERVISORAGENT'S CONTEXT WINDOW

To make informed decisions, the SUPERVISORAGENT is provided with a rich, real-time snapshot of the MAS's state, which we formalize as the *context window*.

---

**Definition 2 (Context Window) .** *The standard context window, $\mathcal{W}$, is a tuple of five key elements:*

$$\mathcal{W} = (N, Q_g, Q_l, T_l, S),$$

*where $N$ is the name of the agent under review, $Q_g$ and $Q_l$ are the global and local tasks, $T_l$ is the* ***local trace*** *of agent $N$'s recent actions and observation summaries, and $S$ is a summary of the agent's latest interaction step. For diagnosing system-wide inefficiencies, we augment this to an extended context window $\mathcal{W}_{ext} = \mathcal{W} \cup \{T_g\}$, where $T_g$ is the* ***global trace*** *of all agent interactions.*

---

## 3.3 THE SUPERVISORAGENT'S ACTION SPACE

The role of the SUPERVISORAGENT is to diagnose high-risk interactions and execute a targeted intervention (Figure 2c). We define three primary intervention contexts, $c \in \mathcal{C} = \{c_{\text{error}}, c_{\text{inefficient}}, c_{\text{excessive}}\}$, which activate one of three core supervision strategies:

- **Proactive Error Correction:** Triggered by $c_{\text{error}}$, this strategy aims to diagnose the root cause of an explicit error and provide a direct fix or a verification task to resolve it.
- **Guidance for Inefficiency:** Triggered by $c_{\text{inefficient}}$, this strategy provides pragmatic, course-correcting hints for sub-optimal behaviors, while also critically permitting productive, albeit repetitive, processes to continue via an *approve* action.
- **Adaptive Observation Purification:** Triggered by $c_{\text{excessive}}$, this strategy refines excessively long or noisy observations to improve the signal-to-noise ratio for the agent.

These strategies are implemented by selecting an action $a$ from the global action space $\mathcal{A}$. The specific subset of permissible actions, $\mathcal{A}(c)$, is formally defined by the intervention context as follows:

$$\mathcal{A}(c) = \begin{cases} \{correct\_observation, provide\_guidance, run\_verification\} & \text{if } c = c_{\text{error}} \\ \{approve, provide\_guidance\} & \text{if } c = c_{\text{inefficient}} \\ \{correct\_observation\} & \text{if } c = c_{\text{excessive}} \end{cases}$$

The implementation of each action is detailed in Section 4.3.

## 4 METHODOLOGY

Building upon the formalism of a Supervised Multi-Agent System (SMAS) introduced in Section 3, we now detail the architecture and operational workflow of our SUPERVISORAGENT (illustrated in Figure 2). Our methodology is structured around three fundamental questions: **What** to supervise, **When** to supervise, and **How** to supervise. We defer the specific implementation details, including all hyperparameters and prompts, to Appendix A.3 and A.7.

### 4.1 WHAT TO SUPERVISE: HIGH-RISK INTERACTION POINTS

The primary targets for our supervision are the three high-risk interaction points defined in our preliminary formalism (Section 3.1, see also Figure 2a): Agent-Agent, Agent-Tool, and Agent-Memory interactions. These points are the primary channels through which errors and inefficiencies are introduced and propagated throughout the system. Our goal is to monitor these specific channels to maintain the operational integrity of the MAS.

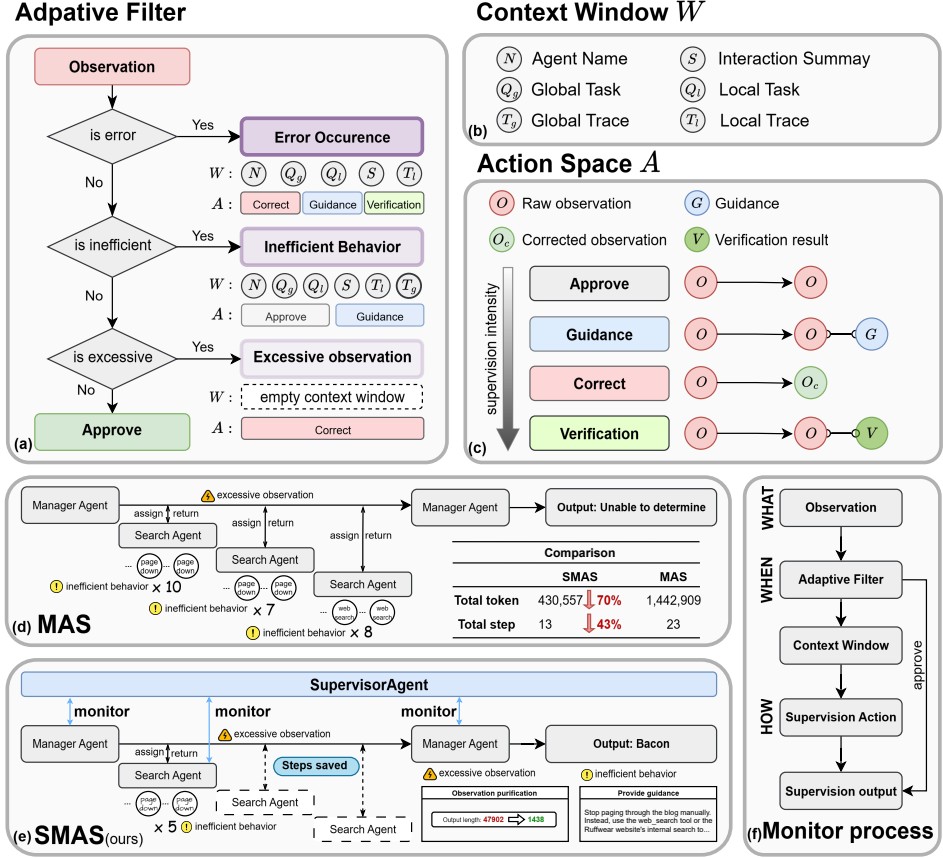

Figure 2: **The architecture and workflow of SUPERVISORAGENT. (a)** The LLM-free adaptive filter for identifying high-risk interactions. **(b)** The context window, aggregating goals and traces for situational awareness. **(c)** The spectrum of intervention actions, from simple approval to intensive verification. **(d, e)** Case study on a GAIA task, comparing the baseline MAS (d) with our SMAS (e), which cuts steps by 43% and token cost by over 70%. **(f)** The supervise workflow for an interaction, from filtering to a final supervision action.

## 4.2 WHEN TO SUPERVISE: THE ADAPTIVE FILTER

While a naive approach might monitor every interaction, the associated computational cost is prohibitive and would undermine our goal of improving efficiency. Therefore, the cornerstone of our framework is a lightweight, LLM-free **adaptive filter** (see in Figure2a) designed to trigger supervision only at critical junctures (see case studies in Figures 2d and 2e). This approach ensures that the SUPERVISORAGENT's resources are deployed judiciously, maximizing impact while minimizing overhead. The filter is designed to be fast and heuristic-based, monitoring the MAS for three pre-defined, high-risk scenarios:

- **Error occurrence:** The manifestation of an explicit error (e.g., in tool use or code execution) is a critical trigger. Unlike current MAS that often pass the full error log into a cluttered context for a subsequent agent to debug, our filter immediately flags these events for a focused, real-time intervention.
- **Inefficient behavior:** An agent may enter a loop of sub-optimal or repetitive actions that, while not explicit errors, lead to high token consumption and latency. Our filter is designed to detect such patterns, such as an agent repeatedly using the page_down action instead of a more direct search strategy.
- **Excessive observation length:** Interactions with tools can return excessively long and noisy observations (e.g., raw HTML) that inflate costs and distract the agent. Our filter identifies such cases for immediate information purification.

### 4.3 How to Supervise: Memory-Augmented, Multi-Level Intervention

Once a high-risk interaction is flagged, SUPERVISORAGENT leverages a rich context window and a spectrum of intervention strategies to deliver a nuanced, effective response.

**Memory-augmented context window.**   To make an effective decision, a supervisor must possess a more comprehensive understanding of the system's state than any single agent. This is why SUPERVISORAGENT is conceptualized with its own memory module, not a simple monitor. As illustrated in Figure 2b, this is achieved through a dynamic **context window** $\mathcal{W}$, which aggregates the global task $Q_g$, the agent's local task $Q_l$, interaction summary $S$, and its recent local action trace $T_l$. Crucially, for diagnosing complex inefficiencies, SUPERVISORAGENT also accesses the **global trace** $T_g$, granting it a holistic perspective that transcends the limited view of any individual agent. This elevated viewpoint is what enables it to provide genuinely strategic guidance.

**A spectrum of intervention actions.**   With this rich context, SUPERVISORAGENT selects an action from a multi-level action space $\mathcal{A}$, adapting intervention intensity tailored to issue severity (Figure 2c). These actions range from a minimal nudge to a comprehensive correction:

- *approve*: A minimal intervention that permits a productive, albeit repetitive, agent behavior to continue. Primarily used in the *inefficient* context, its purpose is to avoid disrupting a process that is pragmatically the best path forward from its current state.
- *provide_guidance*: A semi-intrusive action that steers an agent away from a sub-optimal strategy or logical flaw. This action appends a concise, directive hint to the existing observation, correcting the agent's reasoning path without altering the core context data.
- *correct_observation*: A direct and forceful intervention that refines the agent's sensory input. It is the sole action for *excessive observations*, where it purifies the content, and is also used in *error* contexts to fix factually incorrect data. This action replaces the original raw observation entirely with a cleaned and corrected version.
- *run_verification*: The deepest intervention, used in complex *error* contexts when internal information is insufficient. It invokes a verification sub-agent for external fact-checking or advanced debugging, returning a definitive, verified result.

## 5 Experiments

### 5.1 Experimental Setup

We empirically validate the effectiveness of SUPERVISORAGENT through a series of extensive experiments. We begin by outlining our evaluation metrics, datasets, and baselines. For a more detailed description of the experimental settings, please refer to Appendix A.2.

**Datasets.**   We evaluate our method on a diverse suite of six benchmarks spanning three domains. Our primary benchmark is the challenging GAIA validation set (Mialon et al., 2023), which provides a comprehensive test of an MAS's general problem-solving capabilities. To demonstrate broader applicability, we use five additional benchmarks: for mathematical reasoning, we use AIME 2024 (HuggingFaceH4, 2024) and a random subset of 600 samples from GSM8k-Hard (Gao et al., 2022); for code generation, we use the full HumanEval (Chen et al., 2021) and MBPP (Austin et al., 2021) datasets; and for question answering, we use a subset of 800 samples from the DROP (Dua et al., 2019) dataset, following the sampling strategy of prior work (Zhang et al., 2025c).

**Baselines.**   On several benchmarks, we compare SUPERVISORAGENT against a comprehensive set of agentic systems equipped with web-browsing and code execution capabilities. These baselines fall into two categories: (1) single agent execution methods: including vanilla LLM, Self Consistency CoT (3 answers) (Wang et al., 2023), and CodeAgent (Roucher et al., 2025); and (2) multi-agent systems, including Smolagent (Roucher et al., 2025), OAgents (Zhu et al., 2025), MetaAgent (Zhang et al., 2025h), OWL (role playing) (Hu et al., 2025), and AWorld (Xie et al., 2025; Yu et al., 2025). Detailed descriptions of these baselines are provided in Appendix A.2.2.

Table 1: **Overall performance on the GAIA validation set.** Our SMAS consistently reduces the average token cost comparing to Smolagent baseline while achieving competitive pass@k success rates.

| Method | Avg. Acc. | Avg. Tokens (K) | L1 Acc. | L1 Tokens (K) | L2 Acc. | L2 Tokens (K) | L3 Acc. | L3 Tokens (K) |
|---|---|---|---|---|---|---|---|---|
| CodeAgent | 40.00 | 120.40 | 56.60 | 92.84 | 34.88 | 131.90 | 23.08 | 138.54 |
| OWL | 45.40 | 111.07 | 56.56 | 67.72 | 43.02 | 110.36 | 29.16 | 209.34 |
| OAgents | 49.09 | 340.50 | 66.04 | 260.27 | 47.67 | 358.63 | 19.23 | 444.11 |
| Smolagent | 50.91 | 527.76 | 62.26 | 298.51 | 53.49 | 619.59 | 19.23 | 691.33 |
| AWorld | 60.00 | 128.27 | 67.92 | 69.61 | 62.79 | 164.08 | 34.62 | 133.65 |
| **pass@1** | | | | | | | | |
| Smolagent | 50.91 | 527.76 | 62.26 | 298.51 | 53.49 | 619.59 | 19.23 | 691.33 |
| + **SMAS (ours)** | 50.91 | 371.12 ↓29.68% | 62.26 | 258.28 ↓13.48% | 51.16 | 404.96 ↓34.64% | 26.92 ↑7.69% | 489.22 ↓29.23% |
| **pass@2** | | | | | | | | |
| Smolagent | 58.18 | 467.19 | 69.81 | 275.85 | 59.30 | 548.02 | 30.77 | 589.92 |
| + **SMAS (ours)** | 58.79 ↑0.61% | 389.54 ↓16.62% | 73.58 ↑3.77% | 270.07 ↓2.10% | 56.98 | 420.97 ↓23.18% | 34.62 ↑3.85% | 529.20 ↓10.29% |
| **pass@3** | | | | | | | | |
| Smolagent | 61.82 | 502.40 | 71.70 | 282.14 | 63.95 | 605.05 | 34.62 | 611.87 |
| + **SMAS (ours)** | 63.03 ↑1.21% | 369.52 ↓26.45% | 75.47 ↑3.77% | 276.84 ↓1.88% | 62.79 | 409.05 ↓32.39% | 38.46 ↑3.84% | 427.72 ↓30.10% |

**Implementation details.** To assess model-agnosticism, we test SUPERVISORAGENT with multiple foundation models. For the demanding GAIA benchmark, we primarily use GPT-4.1 as the base model for all agents, and evaluate SUPERVISORAGENT when powered by GPT-4.1 (OpenAI, 2025), Gemini-2.5-pro-0605 (Team, 2025a), and Qwen3-235B-2507 (Team, 2025b). For all other benchmarks, we employ the efficient and powerful Qwen3-32B (Team, 2025b) for both the base agents and the SUPERVISORAGENT to assess performance in a more resource-constrained setting.

**Testbed selection.** We selected Smolagent as our primary experimental testbed, which provides a flexible framework upon which we build our agentic systems (SMAS). Critically, Smolagent's capabilities stem primarily from its internal agentic interactions rather than powerful external tools(e.g. web APIs or solvers). This provides an ideal, controlled environment to isolate and evaluate the direct impact of our SUPERVISORAGENT on an agent's core reasoning and communication processes.

**Metrics.** For GAIA and the code generation benchmarks, we report the standard pass@k metric. For our main baseline, Smolagent, we report pass@1, 2, and 3. For math reasoning, we report the final solve rate (%). For question answering, we report the F1 score for DROP. In all experiments, we meticulously track and report the total token consumption as a primary measure of efficiency.

## 5.2 RESULTS AND ANALYSIS

**Significant efficiency gains with competitive accuracy.** The main experimental results, presented in Table 1, confirm the substantial benefits of SUPERVISORAGENT. On the GAIA validation set, when integrated with the Smolagent framework, SUPERVISORAGENT achieves an average token reduction of **29.68%** at pass@1, while maintaining a statistically equivalent success rate. Notably, the efficiency gains are even more pronounced on more difficult tasks, with token savings reaching **32.39%** on Level 2 and **30.10%** on Level 3 tasks at pass@3.

*Across the other five benchmarks,* SUPERVISORAGENT *generally achieves a Pareto improvement* (see Table 2). In mathematical reasoning, it raises the AIME solve rate by **6.67%** while cutting token costs by **18.92%**. In code generation, it maintains competitive accuracy on HumanEval and further reduces token use by **23.74%**, likely due to its ability to streamline repetitive debugging cycles. Occasionally, SUPERVISORAGENT may overcompress long contexts during purification, causing minor accuracy or F1 drops on certain benchmarks. These results underscore SUPERVISORAGENT's ability to act as a universal efficiency enhancer across diverse problem domains.

**Model-Agnostic generalization.** To demonstrate that the benefits of SUPERVISORAGENT are architectural rather than model-specific, we evaluated it with three different powerful LLMs as its inference engine on GAIA. As shown in Figure 4b, SUPERVISORAGENT *consistently yields significant token savings and maintains robust performance across all models, including GPT-4.1, Gemini-2.5-pro, and Qwen3-235B.* This validates that our supervision framework is a model-agnostic component that can enhance a wide variety of LLM-powered agent systems.

Table 2: **Generalization across diverse benchmarks.** SUPERVISORAGENT consistently reduces token costs while maintaining or improving accuracy on tasks spanning mathematical reasoning, code generation, and question answering. All reported gains are relative to the Smolagent baseline.

| Method | Metrics | GSM-hard | AIME | HumanEval | MBPP | DROP |
|---|---|---|---|---|---|---|
| Vanilla | Acc / F1 (%) | 67.17 | 26.67 | 76.82 | 80.09 | 76.36 |
| | Avg. Tokens (K) | 0.37 | 2.01 | 0.28 | 0.27 | 0.46 |
| CoT SC (3-shot) | Acc / F1 (%) | 69.01 | 30.00 | 77.78 | 81.26 | 77.72 |
| | Avg. Tokens (K) | 2.62 | 14.26 | 1.42 | 1.29 | 2.73 |
| OWL | Acc / F1 (%) | 72.48 | 33.33 | 90.74 | 79.08 | 79.85 |
| | Avg. Tokens (K) | 15.67 | 56.11 | 31.87 | 54.80 | 11.47 |
| MetaAgent | Acc / F1 (%) | 72.14 | 26.67 | 74.08 | 79.86 | 78.16 |
| | Avg. Tokens (K) | 4.35 | 6.24 | 2.59 | 6.39 | 1.43 |
| Smolagent | Acc / F1 (%) | 74.33 | 30.00 | 92.07 | 85.68 | 81.08 |
| | Avg. Tokens (K) | 11.59 | 59.14 | 40.91 | 111.07 | 12.01 |
| **+ SMAS (ours)** | Acc / F1 (%) | 75.50 | 36.67 | 92.68 | 84.43 | 79.80 |
| | Avg. Tokens (K) | 10.55 ↓8.92% | 47.95 ↓18.92% | 31.19 ↓23.74% | 103.71 ↓6.62% | 11.34 ↓5.60% |

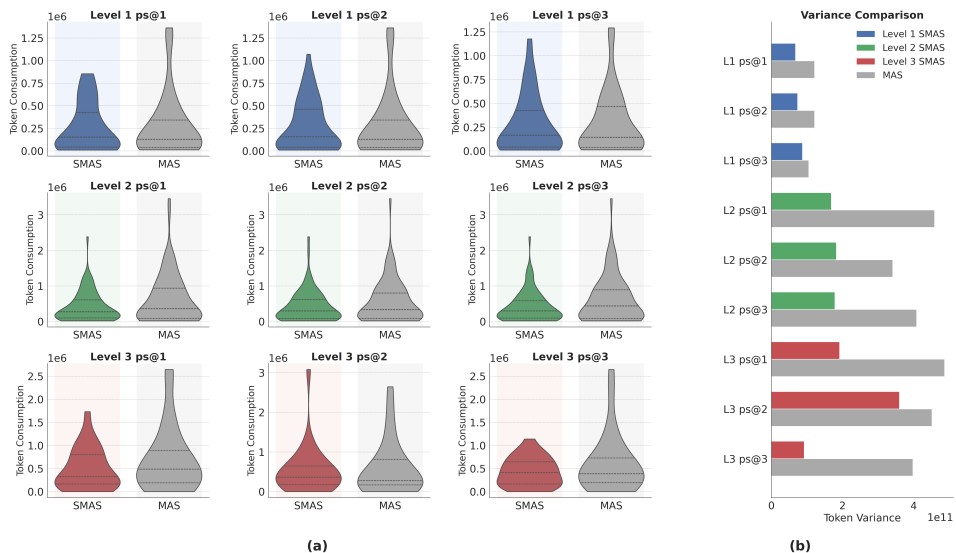

(a)                                                                                      (b)

Figure 3: **SUPERVISORAGENT enhances performance consistency on the GAIA benchmark. (a)** Violin plots of token cost distributions, revealing the more compact and predictable performance of our Supervised MAS (SMAS). **(b)** A direct comparison quantifying the substantial reduction in token cost variance achieved by our SMAS across all difficulty levels.

**Improving robustness and performance consistency.** Beyond average performance, we define robustness as the consistency of an agent's performance. As illustrated by the violin plots in Figure 3, SUPERVISORAGENT *significantly reduces the variance in token consumption per task.* The distributions for the SMAS are visibly shorter and wider, indicating a more concentrated and predictable performance profile. The bar chart on the right further quantifies this, showing a marked decrease in token cost variance, especially for the more complex Level 2 and 3 tasks. This demonstrates that *our method not only makes the MAS more efficient on average but also more reliable and less prone to extreme resource consumption outliers.*

**Ablation study.** We conducted an ablation study on the full GAIA validation set to isolate the impact of SUPERVISORAGENT's three core strategies (Table 3, Figure 4a). A comparison of the full framework with **w/o Correction** (Proactive Error Correction), **w/o Guidance** (Guidance for Inefficiency), and **w/o Purification** (Adaptive Observation Purification) reveals distinct roles. **Purification** is the primary driver of efficiency; disabling it drastically reduces token savings (from 29.68% to 15.96%). Conversely, removing **Correction** or **Guidance** results in the most significant

Table 3: **Ablation study of SUPERVISORAGENT's components** on the full GAIA validation set.

| Method | Avg. Acc. | Avg. Token | Level 1 Avg. Token | Level 2 Avg. Token | Level 3 Avg. Token |
|---|---|---|---|---|---|
| Smolagent | 50.91 | 527,759 | 298,506 | 619,591 | 691,331 |
| + SMAS (w/o Correction) | 47.88 | 354,226 ↓32.88% | 221,515 ↓25.79% | 363,871 ↓41.27% | 592,852 ↓14.24% |
| + SMAS (w/o Guidance) | 48.48 | 363,644 ↓31.10% | 253,591 ↓15.05% | 419,913 ↓32.23% | 401,861 ↓41.87% |
| + SMAS (w/o Purification) | 49.70 | 443,520 ↓15.96% | 270,058 ↓9.53% | 502,937 ↓18.83% | 600,582 ↓13.13% |
| + SMAS | 50.91 | 371,119 ↓29.68% | 258,279 ↓13.48% | 404,955 ↓34.64% | 489,222 ↓29.23% |

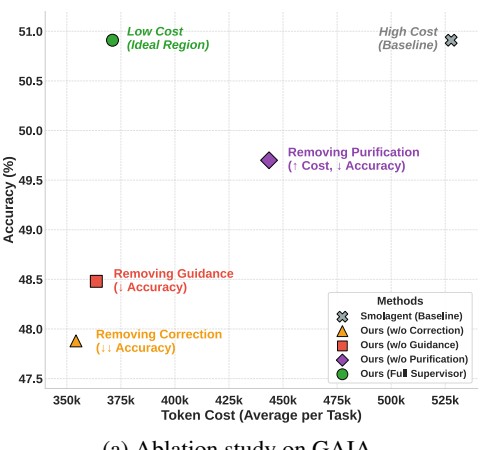

(a) Ablation study on GAIA.

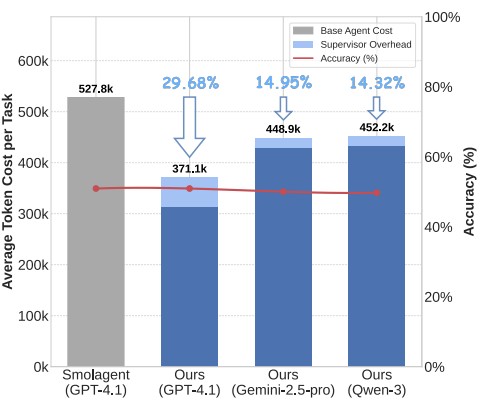

(b) Model Generalization of SUPERVISORAGENT.

Figure 4: **Ablation study and model generalization of SUPERVISORAGENT.** **(a)** Ablation study on challenging GAIA tasks, dissecting the distinct contributions of each module to the framework's overall efficiency and robustness. **(b)** Validation of model-agnosticism, showing that SUPERVISORAGENT consistently delivers token savings across diverse foundation models.

accuracy drops, confirming their necessity for robustness. This underscores a synergistic design: while Purification minimizes cost, Correction and Guidance ensure task success, justifying their marginal overhead. These benefits are particularly pronounced on high-cost tasks (see Appendix A.4.2).

**MAS-Agnostic generalization.** To verify MAS-agnostic feature of SUPERVISORAGENT, we integrate SUPERVISORAGENT into two distinct multi-agent system frameworks: AWorld (Xie et al., 2025) and OAgents (Zhu et al., 2025), and evaluate their performance on the subset of GAIA benchmark(top-10 most token-intensive tasks per GAIA level). The results, presented in Table 4, indicate that SUPERVISORAGENT consistently enhances the performance of both frameworks, underscoring its versatility and effectiveness across different MAS architectures.

Specifically, integrated with AWorld (Xie et al., 2025), our SMAS(AWorld) achieved superior average accuracy over both the original AWorld (without Guard) and the Guard-enabled version. Furthermore, SMAS(AWorld) demonstrated substantial token efficiency, saving **36.54%** on average versus AWorld (with Guard). With savings reaching **48.38%** on Level 3 tasks, it confirms SMAS's ability to enhance tool-intensive MAS. Applying SUPERVISORAGENT to OAgents (Zhu et al., 2025) further validated its general applicability, reducing average token consumption by **39.36%** while maintaining competitive accuracy. Interestingly, the largest token reduction (**50.19%**) occurred on Level 1 tasks, exceeding the savings on Level 3 (**40.63%**). While this might reflect OAgents' inherent proficiency on harder tasks, the significant overall savings underscore SUPERVISORAGENT's broad utility.

**Overhead analysis.** Crucially, all efficiency gains reported in this work represent **net savings**, fully accounting for the cost of SUPERVISORAGENT. As detailed in Table 6, the supervisor itself incurs a modest overhead, averaging only **15.45%** of total token usage, which validates its lightweight design. Regarding latency, the supervisory interventions introduce an average increase of less than one minute and a half per task (Table 7). We consider this temporal cost a justifiable trade-off given the substantial economic savings achieved in complex multi-agent workflows. A comprehensive overhead analysis is provided in Appendix A.4.1.

Table 4: **Cross-framework performance of SUPERVISORAGENT.** Evaluated on GAIA subset (top-10 most token-intensive tasks per level).

| Method | Avg. Acc. | Avg. Token | Level 1 Avg. Token | Level 2 Avg. Token | Level 3 Avg. Token |
|---|---|---|---|---|---|
| Smolagent | 40.00 | 1,446,526 | 933,013 | 2,037,437 | 1,369,131 |
| + SMAS | 46.67 ↑6.67% | 721,332 ↓50.13% | 522,364 ↓44.01% | 960,694 ↓52.85% | 680,939 ↓50.26% |
| AWorld (without Guard) | 23.33 | 155,239 | 50,851 | 217,332 | 166,500 |
| AWorld (with Guard) | 30.00 | 353,738 | 135,413 | 463,083 | 376,878 |
| **AWorld (with SMAS)** | 36.67 ↑6.67% | 224,480 ↓36.54% | 90,569 ↓33.12% | 355,051 ↓23.33% | 194,561 ↓48.38% |
| OAgents | 46.67 | 530,939 | 430,852 | 359,511 | 802,454 |
| + SMAS | 46.67 | 321,957 ↓39.36% | 214,604 ↓50.19% | 274,875 ↓23.54% | 476,393 ↓40.63% |

## 6 DISCUSSION AND CONCLUSION

**Supervisor as a foundational MAS component.** Our work positions SUPERVISORAGENT as a foundational component for future Multi-Agent Systems, akin to established modules like memory banks and tool-usage frameworks. By providing real-time, adaptive supervision, SUPERVISORAGENT alleviates critical challenges of robustness and efficiency that are pervasive across diverse MAS architectures. Its modular design allows for seamless integration with existing systems, enhancing their performance without necessitating fundamental changes to their core logic. This underscores the potential of supervisory agents as universal enhancers of MASs, capable of elevating both reliability and cost-effectiveness across a wide range of applications.

**Comparison with related supervisory agents.** A related concept is the Guard agent in AWorld (Xie et al., 2025), which is invoked at key steps primarily for factual verification to enhance task accuracy. While valuable, its scope differs significantly from our method. SUPERVISORAGENT adopts a broader objective of improving overall system efficiency and robustness through continuous (albeit adaptively filtered) monitoring and a wider range of interventions, including error correction, inefficiency guidance, and observation purification, complementing the Guard's focus on accuracy.

**Broader insights.** Our work also yields critical insights for the broader field. First, we discovered that seemingly "noisy" information, such as HTML structure and truncation cues, serves as a vital signal for ReAct-style agents. The overly aggressive purification can paradoxically harm performance. This highlights a fundamental trade-off between information density and the preservation of environmental texture. Second, our focus on token cost underscores the need for a more holistic efficiency evaluation for MAS. A comprehensive analysis must also account for the frequency and complexity of external tool API calls, which offload significant burdens from the MAS. This very trade-off informed our choice of Smolagent as a primary testbed - its reliance on internal agentic reasoning, rather than powerful external tools, provided a controlled environment to isolate and evaluate our SUPERVISORAGENT's impact on the interaction process itself.

**Future directions.** These insights inform several promising avenues for future work. First, moving beyond heuristic rules, exploring a learning-based adaptive filter could enable more precise, dynamic control over supervisor invocations. This aligns with the broader goal of developing a self-evolving, memory-augmented version of SUPERVISORAGENT. Second, further research should focus on mitigating the latency overhead introduced by supervisory calls to enhance real-time applicability, alongside creating sophisticated purification techniques that address the "noise-as-signal" trade-off. Finally, developing a universal resource consumption metric for MAS remains a critical open challenge. Ultimately, we posit that incorporating such real-time, meta-level supervision is a foundational component for building the next generation of truly scalable and reliable MAS.

**Conclusion.** In this work, we introduced **SUPERVISORAGENT**, a lightweight and non-intrusive meta-agent framework that enhances the robustness and efficiency of Multi-Agent Systems. Through real-time, adaptive supervision, SUPERVISORAGENT mitigates common failure modes and reduces computational overhead using three core strategies: proactive error correction, pragmatic inefficiency guidance, and adaptive observation purification. Our extensive experiments demonstrate a significant Pareto improvement. On the challenging GAIA benchmark, SUPERVISORAGENT reduces token consumption by an average of 29.68% while maintaining competitive task success rates, a crucial step towards building more practical and scalable agentic systems.

## ACKNOWLEDGEMENTS

This work was supported in part by the National Key Research and Development Program of China (2024YFF0907803), Research Fund for International Scientists of National Natural Science Foundation of China (72350710798), National Natural Science Foundation of China (NSFC) under No. 62576285, 62276230, Research Center for Industries of the Future (RCIF) at Westlake University, and Westlake Education Foundation.

## ETHICS STATEMENT

Our work aims to improve the reliability and efficiency of Multi-Agent Systems, a crucial step for developing practical and beneficial autonomous technologies. We believe that by introducing a mechanism for real-time supervision, our framework provides a paradigm not only for performance optimization but also for enhancing the safety and predictability of future agentic systems. Our research was conducted on publicly available benchmarks, did not involve private user data, and adheres to the ICLR Code of Ethics.

## REPRODUCIBILITY STATEMENT

We are committed to ensuring our work is reproducible. The core architecture and logic of SUPERVISORAGENT are detailed in Section 4, with theoretical formalisms in Section 3. For direct replication, we provide all implementation details and final prompts in Appendix A.3, A.7, and our code is available at https://github.com/LINs-lab/SupervisorAgent. The datasets and metrics used in our extensive experiments (Section 5) are all based on publicly available benchmarks, allowing for direct comparison and validation of our results.

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

## CONTENTS

# A APPENDIX

## A.1 LLM USAGE

The large language model (LLM) was utilized as a writing assistant during the preparation of this manuscript. Its application was strictly limited to improving the clarity and grammatical accuracy of the text. Specific uses included rephrasing sentences for better flow and translating initial concepts and drafts from Chinese to English. All core scientific contributions, including the conceptualization of our SUPERVISORAGENT framework, the design of the methodology and experiments, and the analysis and interpretation of the results, are solely the work of the authors. The authors take full responsibility for all claims and the final content of this paper.

## A.2 EXPERIMENTAL SETUP

### A.2.1 DATASETS

Here, we provide a detailed introduction to the datasets used in this paper:

- **GAIA** (Mialon et al., 2023) serves as a benchmark designed to evaluate next-generation LLMs that possess enhanced capabilities through the incorporation of tools, efficient prompting strategies, and access to external search resources. This benchmark comprises over 450 challenging questions, each with a clear and unequivocal answer, necessitating varying degrees of tooling and autonomy for resolution. Accordingly, the questions are categorized into three distinct levels: Level 1 is expected to be solvable by proficient LLMs, while Level 3 signifies a substantial increase in the model's capabilities. Each level includes a fully public development set for validation purposes, as well as a test set containing private answers and associated metadata. In our experiments, we utilize the test set, which encompasses 164 tasks.

- **GSM-hard** (Gao et al., 2022) is an advanced version of the GSM8K mathematics reasoning dataset (Cobbe et al., 2021). This enhanced dataset presents models with increased challenges, featuring larger numerical values and more complex relationships within the problems.

- **AIME-2024** (HuggingFaceH4, 2024) is a dataset comprising problems derived from the American Invitational Mathematics Examination (AIME) 2024. AIME is a prestigious mathematics competition for high school students, recognized for its challenging problems that span various mathematical domains. This benchmark serves multiple purposes: it evaluates the mathematical reasoning capabilities of LLMs, assesses their problem-solving abilities on complex mathematical challenges, and investigates AI performance on structured mathematical tasks.

- **HumanEval** (Chen et al., 2021) is a dataset released by OpenAI that includes 164 programming problems, each containing a function signature, a docstring, a body, and several associated unit tests. These problems were handwritten to ensure that they were not included in the training dataset for code-generation models. This benchmark is crucial for evaluating code-generation models, providing a structured set of challenges in Python that facilitates the assessment of both the quality and correctness of code produced by language models.

- **MBPP**(Mostly Basic Python Problems Dataset) (Austin et al., 2021) comprises approximately 1,000 crowd-sourced Python programming problems that are specifically designed to be solvable by entry-level programmers. The dataset covers essential programming fundamentals and standard library functionalities. Each problem includes a task description, a corresponding code solution, and three automated test cases.

- **DROP**(Data Retrieval Open Answering) Dua et al. (2019) is a reading comprehension benchmark that requires discrete reasoning over paragraphs. This dataset consists of 96,000 questions developed through crowd sourcing and adversarial methods. It challenges systems to resolve references within the questions, which may point to multiple input positions. The tasks entail performing discrete operations, such as addition, counting, and sorting, necessitating a substantially more comprehensive understanding of paragraph content than that demanded by prior datasets. In our experiment, we sampled 800 tasks for evaluation.

### A.2.2 BASELINES

- **Vanilla** is the original Large Language Model (LLM) that processes input using only the question and a basic prompt, without any prompt engineering or external tool integration. This straightforward approach emphasizes the model's inherent capabilities in handling natural language tasks. By operating in this simplistic manner, Vanilla LLM serves as a critical baseline for evaluating the performance of more advanced techniques that incorporate sophisticated prompt strategies or additional tools, thereby providing valuable insights into the effectiveness of various methodologies in natural language processing.

- **CoT-SC**(Chain-of-Thought Self-Consistency) (Wang et al., 2023) serves as a baseline for enhancing the reasoning capabilities of language models. This approach generates multiple reasoning chains, which are then aggregated to produce a coherent summary. By leveraging self-consistency, CoT-SC improves the reliability of the model's outputs, allowing for better performance in complex reasoning tasks. This structured process facilitates deeper analysis of the model's thought processes, providing a foundation for comparing more advanced reasoning strategies and understanding their impact on overall performance.

- **MetaAgent** (Zhang et al., 2025h) is a groundbreaking framework designed to automatically construct multi-agent systems by specifying the objectives of a given task. A distinctive feature of MetaAgent is its ability to generate these multi-agent systems without relying on external training data. This capability allows the produced multi-agent systems to effectively address all scenarios within the specified task domain. The underlying architecture of the Multi-Agent System is based on Finite State Machines(FSM), which facilitates structured decision-making and state transitions, thereby enhancing the system's operational efficiency and adaptability.

- **OWL**(Open Web Language) (Hu et al., 2025) serves as a foundational framework for knowledge representation in multi-agent systems. By enabling agents to process and reason over complex data in a machine-readable format, OWL is crucial for facilitating interoperability among diverse agents. It allows for the creation of ontologies that define intricate relationships and constraints within the environment, thereby enhancing collaborative behaviors among agents. The expressive power of OWL supports advanced inference capabilities, empowering agents to share knowledge effectively and make informed decisions. This framework establishes a robust baseline for evaluating and enhancing the performance of multi-agent systems in various applications.

- **Smolagent** (Roucher et al., 2025) is a lightweight library designed to facilitate the development and implementation of AI agents that can think and operate using code. It emphasizes simplicity and efficiency, enabling users to create multi-agent systems with minimal code. Smolagent's architecture allows for smart threading, dependency management, and context sharing, making it ideal for orchestrating complex tasks. By providing a streamlined framework, Smolagent serves as a foundational model for evaluating the performance and capabilities of more advanced agent-based systems in various applications.

- **OAgents** (Zhu et al., 2025) is a modular multi-agent framework that conducts a thorough empirical study of key agent components (planning, memory, tool use, test-time scaling) on benchmarks such as GAIA and BrowseComp. It delivers great performance among open-source agent frameworks. Importantly, OAgents builds on the lightweight agentience model provided by Smolagent (which emphasises code-based agent orchestration and minimal overhead) and extends it with fine-grained task decomposition, dynamic workflow adaptation, multi-source web browsing and more extensive tool and memory modules.

- **AWorld** (Yu et al., 2025) is an open-source framework for large-scale agent–environment interaction, designed to operationalize the "learning from practice" paradigm in agentic AI. It features a hierarchical multi-agent architecture composed of specialized agents such as the *Execution Agent*, which performs primary reasoning and tool-use operations, and the *Guard Agent*, which intervenes at critical steps to verify and refine intermediate outcomes. AWorld adopts a modular design supporting dynamic supervision, context tracking, and distributed orchestration, enabling efficient coordination across diverse tasks and environments. By treating agents and tools as interchangeable components within a unified orchestration layer, it facilitates flexible composition, concurrent execution, and fine-grained control over reasoning workflows, illustrating a scalable and extensible paradigm for constructing adaptive multi-agent systems.

Table 5: Hyperparameter settings for the Heuristic-Based Adaptive Filter across different benchmarks. The symbols correspond to the definitions in Algorithm 1.

| Condition | Parameter (Symbol) | GAIA | HumanEval | MBPP | AIME | DROP | GSM-Hard |
|---|---|---|---|---|---|---|---|
| *Inefficient* | Step Check Interval ($\tau_{\text{step}}$) | 8 | 6 | 4 | 4 | 4 | 4 |
| | Loop Detection Window ($\tau_{\text{loop}}$) | 5 | 5 | 3 | 3 | 3 | 3 |
| *Excessive* | Length Threshold ($\tau_{\text{len}}$) | | | 3000 | | | |

## A.3 IMPLEMENTATION DETAILS

In this section, we provide a detailed description of how the conceptual framework of SUPER-VISORAGENT is implemented in our codebase. Our implementation is centered around the `supervise_and_correct` function, which serves as the primary entry point for all supervisory actions. We structure our explanation following the same *What, When, and How* logic presented in our main methodology.

### A.3.1 WHAT TO SUPERVISE: THE ACTIONSTEP OBJECT

Our supervision targets the discrete interaction steps performed by each agent within the MAS. In our framework, every such interaction is encapsulated in a data structure we refer to as an `ActionStep` object. This object contains all relevant information for a single step, including the agent's thought process (`model_output`), the executed `tool_calls`, the resulting `observations`, and an `error` attribute which is populated if an exception occurs. Our SUPERVISORAGENT is implemented as a callback function that intercepts every `ActionStep` object generated by any agent in the system.

### A.3.2 WHEN TO SUPERVISE: THE PRIORITIZED ADAPTIVE FILTER

To avoid the prohibitive cost of constant intervention, we employ a lightweight, LLM-free adaptive filter. This filter is implemented as a prioritized conditional chain at the beginning of the `supervise_and_correct` function. It evaluates each `ActionStep` to determine if supervision is warranted. The conditions are checked in the following order of precedence:

1. **Sub-Agent Completion**: The highest priority is to check if the observation contains a final report from a sub-agent (identified by the presence of a `"<summary_of_work>"` string). If so, it triggers the specialized `Adaptive Observation Purification` strategy to distill the findings for the manager agent.
2. **Error Occurrence**: If the `step.error` attribute is not `None`, the `Proactive Error Correction` strategy is triggered. Our implementation includes a defensive check to ensure this does not fire for known, non-critical tool failures that the base agent can handle.
3. **Inefficient Behavior**: If no error is present, we then check for inefficiency using our heuristic-based `_check_for_inefficiency` function. This function detects patterns such as hard loops (identical actions and observations) and excessive step counts for a given sub-task, triggering the `Guidance for Inefficiency` strategy.
4. **Excessive Observation Length**: Finally, if none of the above conditions are met, the filter checks if the length of the `step.observations` string exceeds a pre-defined threshold $\tau_{\text{len}}$ (3,000 characters in our implementation). If it does, the general type of `Adaptive Observation Purification` strategy is activated.

If none of these trigger conditions are met, the step is approved by default, thereby avoiding any unnecessary LLM-based supervision overhead. The filter's sensitivity is governed by three key hyperparameters: $\tau_{\text{step}}$ and $\tau_{\text{loop}}$ modulate the detection of inefficient behaviors, while $\tau_{\text{len}}$ defines the threshold for identifying excessive observations. The complete logic of this heuristic-based mechanism is formalized in Algorithm 1.

**Sensitivity analysis and hyperparameter configuration.** Table 5 details the hyperparameter settings for our Heuristic-Based Adaptive Filter across different benchmarks. We conducted a sensitivity analysis on the excessive observation length threshold ($\tau_{\text{len}}$) using the representative subset

---

**Algorithm 1** Heuristic-Based Adaptive Filter Logic

---

**Input:** Current execution history $\mathcal{H}$ (sequence of steps), Current observation $o$, Error status $e$
**Require:** Hyperparameters: $\tau_{\text{step}}$ (step check interval), $\tau_{\text{loop}}$ (loop detection window), $\tau_{\text{len}}$ (length threshold)
**Output:** Boolean flag $trigger$, Intervention context $c$
  1: $trigger \leftarrow$ False
  2: $c \leftarrow$ None
                                       ▷ *Priority 1: Check for explicit runtime errors*
  3: **if** $e$ is **True then**
  4:     **return** (True, $c_{\text{error}}$)
  5: **end if**
                                    ▷ *Priority 2: Check for inefficient behaviors*
  6: $N \leftarrow \text{Length}(\mathcal{H})$
  7: **if** $N > 0$ **and** $N \pmod{\tau_{\text{step}}} = 0$ **then**                ▷ Periodic strategy check
  8:     **return** (True, $c_{\text{inefficient}}$)
  9: **end if**
10: **if** $N \geq \tau_{\text{loop}}$ **then**                             ▷ Repetitive loop detection
11:     $\mathcal{A}_{\text{recent}} \leftarrow \text{GetLastToolCalls}(\mathcal{H}, \text{window} = \tau_{\text{loop}})$
12:     **if** $|\text{Unique}(\mathcal{A}_{\text{recent}})| = 1$ **then**
13:         **return** (True, $c_{\text{inefficient}}$)
14:     **end if**
15: **end if**
                                     ▷ *Priority 3: Check for excessive information*
16: **if** $\text{Length}(o) > \tau_{\text{len}}$ **then**
17:     **return** (True, $c_{\text{excessive}}$)
18: **end if**
19: **return** ($trigger$, $c$)

---

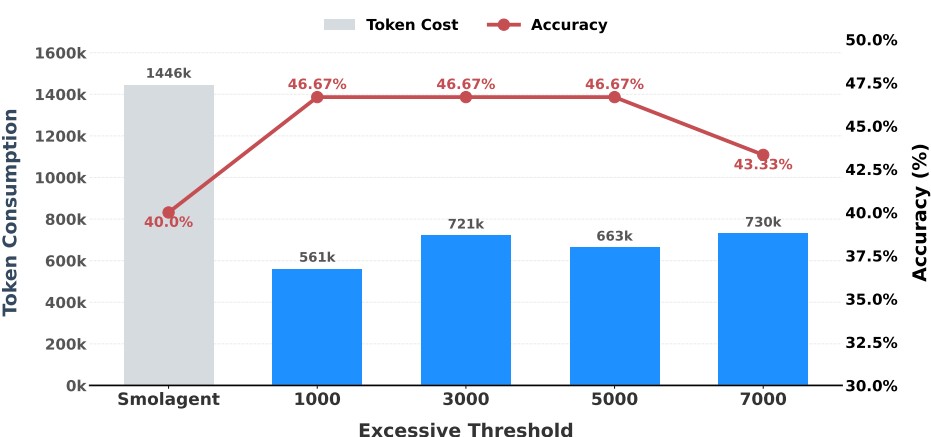

Figure 5: **Sensitivity analysis on excessive threshold ($\tau_{\textbf{len}}$).** Evaluated on GAIA subset (top-10 most token-intensive tasks per level).

of the GAIA benchmark (top-10 most token-intensive tasks per level), as illustrated in Figure 5. The results indicate that our method maintains robust performance across a wide range of threshold values, demonstrating its adaptability without significant degradation. Although slight performance peaks are observed at $\tau_{\text{len}} = 1000$ and $\tau_{\text{len}} = 5000$, we selected 3000 as the default value. This choice strikes a prudent balance between sensitivity (catching enough noise) and specificity (preserving useful context), ensuring optimal performance across diverse tasks while minimizing the risk of over-intervention.

**Configuration for OAgents.** For the OAgents framework (Zhu et al., 2025), we adjusted the parameters to $\tau_{\text{step}} = 6$, $\tau_{\text{loop}} = 3$, and $\tau_{\text{len}} = 10000$. The significantly higher $\tau_{\text{len}}$ (compared to 3000

in Smolagent) is necessitated by OAgents' architecture, which tends to generate extensive verbose outputs due to its complex tool usage and memory retrieval modules. A higher threshold is essential here to effectively identify truly excessive information without triggering false positives on standard OAgents operations.

**Adaptation for AWorld: An MCP-Based Approach.** Unlike the external heuristic filter used in Smolagent, the integration with AWorld (Yu et al., 2025) leverages its native tool-use architecture. We implemented the SUPERVISORAGENT as a **Model Context Protocol (MCP)** service, allowing it to be dynamically discovered and invoked by the AWorld agent. This adaptation involves three key modifications:

- **Mandatory Invocation via System Prompt:** We refined AWorld's system prompt to enforce a protocol where the agent *must* invoke the SUPERVISORAGENT during critical phases—specifically "Information Gathering" and "Thinking Process Reviewing". This ensures the SUPERVISORAGENT acts as a mandatory gatekeeper for logical consistency.
- **Capability-Based Routing:** We defined a specific MCP schema where the SUPERVISORAGENT broadcasts its capabilities, including `Error Root Cause Diagnosis`, `Structured Information Synthesis`, and `Workflow Efficiency Assessment`. This allows the AWorld agent to match its current execution status (e.g., encountering an exception or synthesizing search results) with the appropriate Supervisor function.
- **Trigger Mechanism:** Instead of counting token length, the trigger is semantic. Explicit error returns from AWorld's system serve as triggers for `error_analysis`, while the `sub_agent_result_synthesis` and `inefficiency_analysis` modes are seamlessly integrated into the MCP process flow to facilitate output verification.

### A.3.3 HOW TO SUPERVISE: THE INTERVENTION PIPELINE

Once the adaptive filter flags an interaction, the `supervise_and_correct` function executes a three-stage intervention pipeline:

**1. Context Aggregation** Before making a decision, the Supervisor aggregates a context window ($\mathcal{W}$). This process involves retrieving the global task ($G$) and the agent's local task ($L$), formatting the agent's recent local action history ($T_l$) via the `_format_local_trace_for_prompt` function, and generating a summary of the current step ($S$) using the `_summarize_interaction` function. For inefficient behavior, the full global trace ($T_g$) is also included.

**2. LLM-based Decision Making** The aggregated context is then compiled into a specialized prompt tailored to the triggered supervision type (e.g., `Proactive Error Correction`). This prompt instructs our main model (e.g., GPT-4.1) to analyze the situation and return a structured JSON object containing its `analysis`, a chosen `action` (from the set {`approve`, `correct_observation`, `provide_guidance`, `run_verification`}), and the necessary `parameters` to execute that action.

**3. Action Execution** The returned JSON is parsed, and the chosen action is executed.

- `correct_observation`: The original `step.observations` is entirely replaced with the `new_observation` provided in the parameters. A "[Supervisor's Note: ...]" is prepended to inform the agent of the modification.
- `provide_guidance`: The `guidance` string from the parameters is appended to the end of the existing `step.observations`, leaving the original sensory data intact while providing a corrective hint.
- `run_verification`: The `task` parameter is passed to a dedicated, fully-equipped verification agent, and its conclusive findings are appended to the `step.observations`.

### A.4 EXTENDED EXPERIMENTAL ANALYSIS

### A.4.1 OVERHEAD ANALYSIS OF SUPERVISORAGENT

**Token overhead analysis** The token overhead of SUPERVISORAGENT itself is shown in Table 6 and Figure 6. We analyze the token consumption on the GAIA validation set under different pass@k

Table 6: **Token efficiency analysis on GAIA validation set.** Comparison of token consumption across different pass@k settings.

| Method | Avg. Tokens (K) | L1 Tokens (K) | L2 Tokens (K) | L3 Tokens (K) |
|---|---|---|---|---|
| **pass@1** | | | | |
| Smolagent | 527.76 | 298.51 | 619.59 | 691.33 |
| + Supervised MAS | 314.07 ↓40.49% | 220.63 ↓26.09% | 342.18 ↓44.77% | 411.58 ↓40.47% |
| + Supervised MAS (NET) | 371.12 ↓29.68% | 258.28 ↓13.48% | 404.96 ↓34.64% | 489.22 ↓29.23% |
| **pass@2** | | | | |
| Smolagent | 467.19 | 275.85 | 548.02 | 589.92 |
| + Supervised MAS | 329.51 ↓29.47% | 231.96 ↓15.91% | 354.21 ↓35.37% | 446.64 ↓24.29% |
| + Supervised MAS (NET) | 389.55 ↓16.62% | 270.07 ↓2.10% | 420.97 ↓23.18% | 529.20 ↓10.29% |
| **pass@3** | | | | |
| Smolagent | 502.40 | 282.14 | 605.05 | 611.87 |
| + Supervised MAS | 312.06 ↓37.89% | 236.28 ↓16.25% | 342.36 ↓43.42% | 366.31 ↓40.13% |
| + Supervised MAS (NET) | 369.52 ↓26.45% | 276.84 ↓1.88% | 409.05 ↓32.39% | 427.72 ↓30.10% |

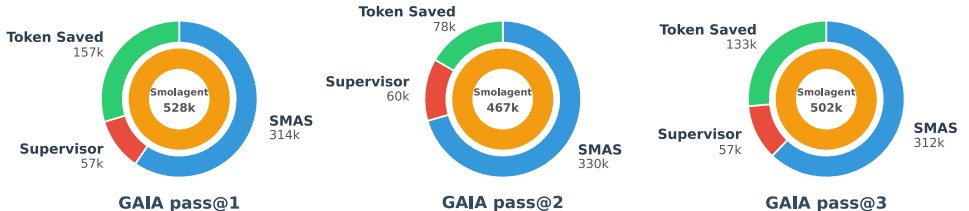

Figure 6: SUPERVISORAGENT **overhead on the GAIA benchmark.**

settings. The results indicate that integrating SUPERVISORAGENT leads to a significant reduction in overall token usage across all complexity levels (L1, L2, L3) and pass@k configurations. Specifically, SUPERVISORAGENT achieves an average token saving of **35.95%** across all settings compared to the baseline Smolagent. This substantial decrease in token consumption highlights SUPERVISORAGENT's effectiveness in optimizing the multi-agent system's efficiency by reducing unnecessary interactions and streamlining the reasoning process.

Notably, even when accounting for the additional tokens introduced by SUPERVISORAGENT's supervisory interventions, the net token consumption remains significantly lower than that of the baseline (already reported in main content). And the token overhead of SUPERVISORAGENT only contains about **15.45%** in average of the total tokens used in the Smolagent baseline. This demonstrates that the benefits of improved efficiency and reduced redundancy far outweigh the costs associated with supervision.

**Latency Overhead Analysis.** Table 7 and Figure 7 analyzes the temporal impact of SUPERVISORAGENT. While integrating the supervisor introduces an average latency increase of **37.27%**, this translates to an absolute delay of less than **1.5 minutes** per task. Crucially, the ablation study reveals that Adaptive Observation Purification is the primary driver of this latency. Notably, the *w/o Purification* variant exhibits a runtime nearly identical to the baseline (236.21s vs. 233.96s). This indicates that the overhead is strictly tied to the processing of excessive information. We consider this a strategic trade-off: exchanging a modest temporal cost for substantial economic (token) savings and enhanced system robustness.

### A.4.2 PERFORMANCE ANALYSIS ON TOKEN-INTENSIVE SCENARIOS

Complementing the comprehensive ablation study on the full GAIA benchmark (Table 3), this section zooms in on the most demanding scenarios: the top-10 most token-intensive tasks per GAIA level. This analysis aims to evaluate the scalability and robustness of SUPERVISORAGENT under extreme computational loads.

Table 7: **Ablation study of SUPERVISORAGENT's components regarding Latency** on the GAIA validation set. Average latency (in seconds) is reported for different complexity levels.

| Method | Avg. Acc. | Avg. Latency (s) | Level 1 Avg. Lat. (s) | Level 2 Avg. Lat. (s) | Level 3 Avg. Lat. (s) |
|---|---|---|---|---|---|
| Smolagent | 50.91 | 233.96 | 155.83 | 247.47 | 348.58 |
| + SMAS (w/o Correction) | 47.88 | 280.77 ↑20.01% | 193.98 ↑24.48% | 276.50 ↑11.73% | 471.81 ↑35.35% |
| + SMAS (w/o Guidance) | 48.48 | 271.95 ↑16.24% | 211.04 ↑35.43% | 291.22 ↑17.68% | 332.38 ↓4.65% |
| + SMAS (w/o Purification) | 49.70 | 236.21 ↑0.96% | 137.00 ↓12.08% | 252.01 ↑1.83% | 386.19 ↑10.79% |
| + SMAS | 50.91 | 321.15 ↑37.27% | 233.11 ↑49.59% | 320.93 ↑29.68% | 501.31 ↑43.81% |

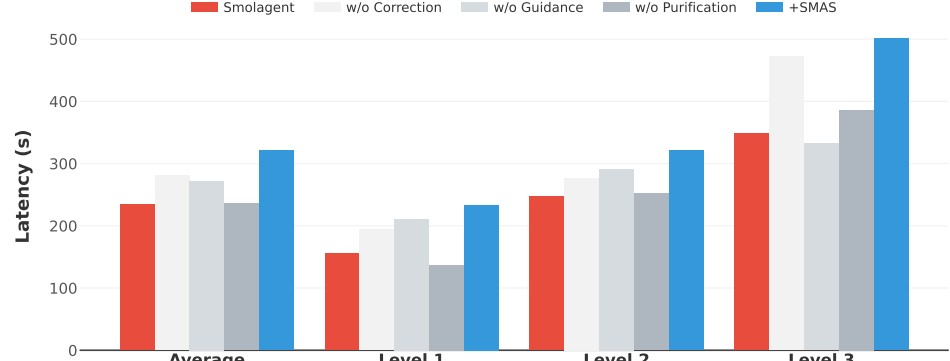

Figure 7: **SUPERVISORAGENT latency on the GAIA benchmark.**

Table 8: **Ablation study of SUPERVISORAGENT's components** on the subset of GAIA benchmark (top-10 most token-intensive tasks per GAIA level).

| Method | Avg. Acc. | Avg. Token | Level 1 Avg. Token | Level 2 Avg. Token | Level 3 Avg. Token |
|---|---|---|---|---|---|
| Smolagent | 40.00 | 1,446,526 | 933,013 | 2,037,437 | 1,369,131 |
| + SMAS (w/o Correction) | 40.00 | 719,075 ↓50.28% | 426,786 ↓54.26% | 755,543 ↓62.91% | 974,895 ↓28.79% |
| + SMAS (w/o Guidance) | 40.00 | 706,831 ↓51.14% | 453,623 ↓51.38% | 913,109 ↓55.18% | 753,761 ↓44.95% |
| + SMAS (w/o Purification) | 46.67↑6.67% | 851,747 ↓41.11% | 585,411 ↓37.26% | 990,769 ↓51.37% | 979,061 ↓28.49% |
| + SMAS | 46.67 ↑6.67% | 721,332 ↓50.13% | 522,364 ↓44.01% | 960,694 ↓52.85% | 680,939 ↓50.26% |

**Amplified Efficiency and Robustness.** As detailed in Table 8, the benefits of SUPERVISORAGENT are significantly amplified in high-complexity regimes. While the average token reduction on the full dataset is 29.68%, SMAS achieves a remarkable **50.13%** reduction on this intensive subset. More importantly, unlike the full dataset where accuracy remains stable, SMAS yields a distinct accuracy improvement of **6.67%** (from 40.00% to 46.67%) on these hard tasks. This suggests that as task complexity and context length increase, the Supervisor's interventions become indispensable not just for cost-saving, but for enabling the agent to complete tasks that were previously intractable due to context overflow or reasoning derailment.

**Component Contribution in Extremes.** The ablation results on this subset reinforce the synergistic roles of our three strategies. The *w/o Purification* variant shows a drastic drop in efficiency (savings drop from 50.13% to roughly 40%), confirming that **Adaptive Observation Purification** is the primary countermeasure against the exponential token growth in complex tasks. Meanwhile, the removal of Correction or Guidance leads to a sharp decline in accuracy back to the baseline level (40.00%), verifying that these modules are the key safety guardrails that allow the system to navigate long-horizon tasks successfully.

## A.5 FAILURE MODE ANALYSIS

While SUPERVISORAGENT demonstrated robustness across benchmarks, it relies on backbone LLMs and is thus subject to their inherent limitations. We identify three primary failure modes and their implications.

**Information Loss during Purification.** The *Adaptive Observation Purification* module faces a trade-off between context reduction and information preservation. In extreme cases observed in the

OAgents framework, where single observations exceeded 200,000 characters, the Supervisor risks hallucinating or omitting critical details during compression. However, our empirical results suggest that the system's resilience to context overflow generally outweighs the cost of granular information loss, as evidenced by the overall token savings and success rates (see in Table 4).

**Ineffective Guidance and Loops.** The *Inefficiency Guidance* module may occasionally provide suboptimal advice or fail to break a stubborn loop. To mitigate the risk of the Supervisor itself becoming a source of latency (e.g., engaging in an infinite correction loop with a non-responsive agent), we enforce a hard constraints of maximum *two* guidance interventions per sub-task. While this design prioritizes bounded latency over guaranteed resolution for the hardest tasks, it effectively prevents runaway costs.

**Variance in Trigger Frequency across Backbones.** Contrary to the assumption that a supervisor acts uniformly, our analysis reveals that SUPERVISORAGENT exhibits different operational behaviors depending on the backbone model's capability. For instance, as shown in our logs with Qwen3-235B, less capable models trigger the *Error Correction* module significantly more often due to frequent basic failures (e.g., malformed JSON tool calls). This frequent firing increases the Supervisor's token overhead, partially explaining the lower net token savings compared to GPT-4.1 (see in Figure 4b). Conversely, stronger models may trigger supervision too rarely if the heuristic filter is not sensitive enough to their subtle logic errors. This highlights that while the *framework* is model-agnostic, the *efficiency gains* are correlated with the backbone LLM's adherence to instruction following.

## A.6 CASE STUDY

---

**GAIA Benchmark Case Information**

**Task ID**: 5b2a14e8-6e59-479c-80e3-4696e8980152

**Level**: 3

**Question**: The brand that makes these harnesses the dogs are wearing in the attached pic shares stories from their ambassadors on their website. What meat is mentioned in the story added Dec 8th 2022?

**Attached iamge**: 5b2a14e8-6e59-479c-80e3-4696e8980152.jpg

---

### Smolagent

**1. Activate first search_agent**:
**Failure attribution**: page_down 10 times and end up searching
**Output of first search_agent**:

```
Observations: ### 1. Task outcome (short version):
Despite a detailed investigation and navigation through the RC Pets
 official website and blog section, I was unable to locate the
ambassador story posted on December 8th, 2022. Therefore, I cannot
report any mention of meat in its content.
```

**2. Activate another search_agent**:
- Plan: turn to access historical versions of the website
- page_down for 7 times
- Then web_search other resources
- Output of search_agent

  ```
  Observations: ### 1. Task outcome (short version):
  Despite a detailed investigation and navigation through the RC
  Pets official website and blog section, I was unable to locate
  the ambassador story posted on December 8th,2022. Therefore, I
  cannot report any mention of meat in its content.
  ```

**3. Activate third search_agent**:

- Plan: Thought: Since all direct website and archive attempts failed, I will extend my search to include RC Pets' official social media (Instagram, Facebook, Twitter) and broader web, as brands often share ambassador stories across platforms and tag or date posts.

- web_search for 8 times

- Output of search_agent

```
Observations: ### 1. Task outcome (short version):
No evidence was found of any RC Pets ambassador or partner story
posted on December 8th, 2022, on official RC Pets Instagram,
Facebook, Twitter, or the broader web. No blog posts, community
group stories, or ambassador reposts regarding RC Pets for that
date or month could be
located, and no relevant content mentioning meat was found.
```

**Original final answer**:

```
Final answer: After an exhaustive investigation across all
available RC Pets channels (official website, blog, and social
media), Wayback Machine, pet influencer blogs, hashtags, and pet
community forums, no ambassador story or repost by RC Pets or
affiliated
ambassadors/partners could be found for December 8th, 2022. No
story mentioning any meat was identified for the date or month, and
 no content relevant to the request is accessible through any
official or community web source. All logical avenues for
retrieving the answer were
attempted and exhausted.
```

**Reformulated answer**: Unable to determine

---

Smolagent + SupervisorAgent

**Supervision type: Basic_extraction** - 8 times
**Supervision type: Inefficiency_analysis** - 2 times

- First time

```
guidance: Providing guidance to agent: 'Stop paging through the
blog manually. Instead, use the web\_search tool or the Ruffwear
website's internal search to find the specific ambassador story
posted on December 8th, 2022. You could search for 'Ruffwear
ambassador story December 8 2022' or check if the blog section
has a date-filter or text search. Once you locate the precise
article, examine it for any mention of meat, noting a quote for
context.'

Analysis: The agent is stuck repeatedly using 'page\_down'
without making meaningful progress toward locating the December 8
th, 2022 story among the 'ambassadors' blog entries. There are 82
 pages, and there's no indication the agent is close to the
correct blog post. This paginated browsing is highly inefficient
when a direct search or filter by date would be much faster. It's
 overwhelmingly more efficient to instead search for the specific
 blog post by date ('December 8th, 2022') using the web\_search
tool or any site search functionality if available.
```

- Second time

```
approve
```

**Supervision type: sub_agent_result_synthesis** - 1 time

- Original output length: 47902

- Length after process: 1438

```
content = """
### Official Ruffwear Ambassador Stories Section
- Ruffwear ambassador stories are shared in 'Explored: A Ruffwear
 Blog' (https://ruffwear.com/blogs/explored/tagged/ambassadors).

### Story Posted December 8, 2022
- Title: "Snow Camping With Theresa & Cassie"
- Author: Theresa Silveyra
- Date Published: December 8, 2022 ("2022-12-08 08:00:01 -0800")
- Story URL: https://ruffwear.com/blogs/explored/snow-camping-
with-theresa-cassie

### Meat Mentioned in the Story
- The only specific meat mentioned is "bacon".

> Specific quoted context from the story:
> "Of course, barely a minute had passed before Cassie's longing
eyes and dangling tongue signaled a desire to play or, perhaps, a
 desire for treats. She led the way back, her fluffy white tail
raised high in excitement, ready to take on this new year and,
more likely, ready to devour the New Year's Day bacon being
cooked up in camp."

### Confirmation of Accuracy
- The article and details are confirmed by both page metadata and
 direct content search.
- Searches for "meat", "chicken", "beef", "turkey", "pork", "fish
", "salami", and "sausage" yielded no other results; only bacon
was present.

### Additional Context
- Author attribution and publication date are validated.
- Other foods or treats are referenced, but bacon is the only
explicit meat in the text.

### Direct Answer
- The meat mentioned in the Ruffwear ambassador story posted on
December 8th, 2022 is bacon.
"""
```

## A.7 PROMPTS

**Base prompt** for SUPERVISORAGENT

```
base_prompt = f"""
Role: You are an expert supervisor in a multi-agent system. Your
role is to monitor an agent's actions, ensure alignment with the
main goal, correct errors, and optimize the workflow.

Objective: The overall objective (Global Task) is: "{global_task}"

Agent context:
You are currently reviewing an action from the agent '{agent_name
}'.
This agent's specific sub-task (Local Task) is: '{local_task}'.
Here is the current local execution trace: {local_trace_str}
```

```
Here is the summary of the agent's latest thought process and the
resulting observation:
    {summary}
    {current_state_summary}

Rules:
1.  Assess Necessity: First, assess if intervention is truly
necessary. If the agent's action and observation are correct and
productive, use the "approve" action. Avoid unnecessary
interventions.
2.  Be Decisive: When an intervention is needed, choose the most
effective action to move the project forward.
3.  Output Format: Your response MUST be a valid JSON object.

Actions:
Your available actions are:
-  'approve': The agent's action is correct and requires no changes
.
-  'correct_observation': The observation contains errors or can be
 significantly improved (e.g., filtered, summarized, extracted).
You will provide a corrected version.
-  'provide_guidance': The observation is correct, but the agent's
thinking or next step is flawed. You will provide a hint or
corrected reasoning to guide the agent.
-  'run_verification': You have doubts about the factual accuracy
of the observation and need an external assistant to verify it.

Your response MUST be a JSON object with the following structure:
{
  "analysis": "Your brief analysis of the situation, explaining
  your reasoning for the chosen action.",
  "action": "ONE of the available actions: ['approve', '
  correct_observation', 'provide_guidance', 'run_verification']",
  "parameters": {
      "new_observation": "IF action is 'correct_observation',
      provide the refined observation here.",
      "guidance": "IF action is 'provide_guidance', provide a clear
       hint or instruction for the agent's next thought process.",
      "task": "IF action is 'run_verification', provide the
      verification question for the assistant."
  }
}"""
```

### Prompt for "error_occurrence"

**Base Prompt:**

...

**Addtional Prompt:**

```
f"""
**Role**:You are an expert Debugger and AI Diagnostician. Your
primary goal is to understand the root cause of an error and
provide the most effective solution to get the agent back on track.

**Situation**:
The agent's last action resulted in a critical error, which is
detailed in the "summary" of the agent's action below. **Approval
is not an option; you must intervene.**

**--- Your Debugging Framework (MANDATORY) ---**
```

```
Before generating your JSON output, you MUST follow this structured
 thinking process:

**Step 1: Analyze the Error**
    - What is the precise error message and type (e.g., 'Tool Error
    ', 'Python Exception', 'APIError')?

**Step 2: Examine the Context**
    - Review the 'local_execution_trace' and the agent's 'thought'
    process leading to the error.
    - What was the agent *trying* to accomplish?
    - Was the tool call or code it executed ('summary' section)
    syntactically correct but logically flawed?

 **Step 3: Root Cause Diagnosis**
    - Based on the error and context, what is the single most
    likely root cause?
    - (e.g., "The agent passed a natural language string to a tool
    expecting a mathematical expression.", "The agent is trying to
    access a file that does not exist.")

**Step 4: Formulate a Solution Strategy**
    - Based on the root cause, determine the best intervention:
        - If the error can be fixed by correcting the agent's **
        next thought process or action**, choose 'provide_guidance
        '. This is the most common case for logical errors.
        - If the error was caused by faulty information in the **
        previous observation** that the agent is now acting upon,
        choose 'correct_observation'.
        - If you lack critical information to solve the error and
        need to consult an external source, choose '
        run_verification'.

**--- YOUR ACTIONABLE OUTPUT (JSON) ---**
Based on your diagnosis, provide your final decision in the JSON
format.
```

### Prompt for "inefficiency_behavior"

**Base Prompt:**

...

**Additional Prompt:**

```
f"""
**Role**:You are a pragmatic and experienced AI workflow strategist
. Your primary goal is to ensure the agent team achieves its task
in the most efficient way **from its current state**.

**Situation**:An inefficiency trigger has been activated for agent
'{agent_name}'. **This is a flag for you to review, NOT a
confirmation of a problem.** The agent might be engaged in a
necessary, methodical process.

**Global Execution Trace**:
{global_trace_str}

**--- Your Decision Framework (MANDATORY) ---**
Before generating your JSON output, you MUST follow this structured
 thinking process:
```

```
**Step 1: Goal & Plan Inference**
    - Based on the 'Global Execution Trace', what is the agent's
    immediate, implicit plan?
    - (e.g., "The agent is clearly trying to collect all rows of a
    data table by repeatedly using 'page_down'.")

**Step 2: Progress Assessment**
    - Is the agent making tangible progress towards its inferred
    goal?
    - Is each new step yielding new, relevant information (even if
    it's just more rows of the same table)?
    - How close is the agent to completing this sub-task? (e.g., "
    It is on page 10 of 13, it is very close to getting all the
    data.")

**Step 3: Cost-Benefit Analysis of Intervention**
    - **Compare two costs**:
        - **Cost A**: The estimated cost (time, tokens) of letting
        the agent **continue** its current, perhaps clumsy, path to
         completion.
        - **Cost B**: The estimated cost of **interrupting** the
        agent, guiding it to a new path, and having it **start over
        ** on that new path.
        - **CRITICAL QUESTION**: Is the agent "one step away" from
        solving its sub-task? If so, interrupting it is almost
        always the wrong decision, even if a theoretically "better"
         path exists.

**Step 4: Decision and Justification**
    - Based on the analysis above, decide between 'approve' and '
    provide_guidance'.

**--- YOUR ACTIONABLE OUTPUT (JSON) ---**
You must choose ONE of the following two actions:

**1. If you decide the agent should continue:**
    - **Condition**: The agent is making clear, incremental
    progress AND is close to completing its sub-task (Cost A < Cost
     B).
    - **Action**: MUST be '"approve"'.
    - **Analysis**: Briefly explain *why* the agent's current path,
     while perhaps repetitive, is the most pragmatic way forward
    from its current state. (e.g., "The agent is methodically
    paginating through a table to gather all data. Although
    repetitive, this is a valid and necessary process. It is on
    page 10 of 13 and about to succeed. Intervention would be
    disruptive.")

**2. If you decide the agent is truly stuck:**
    - **Condition**: The agent is in a non-productive loop (e.g.,
    getting the same observation repeatedly) OR the alternative
    path is overwhelmingly more efficient and the agent is not
    close to finishing (Cost B << Cost A).
    - **Action**: MUST be '"provide_guidance"'.
    - **Analysis**: Briefly explain the root cause of the
    inefficiency.
    - **Guidance**: The 'guidance' parameter MUST contain a clear,
    concrete, and actionable instruction that represents a *
    significantly* better strategy. (e.g., "Instead of scrolling,
    use the 'web_search' tool with the query 'who had the most BB
    for the 1977 Yankees' to get the answer directly.")
"""
```

---

**Prompt for "excessive observation length"**

```
f"""
# Role: AI Agent Observation Compressor
You are a specialized data compression model for an AI agent. Your
sole purpose is to process raw observations (HTML, text, etc.) and
reduce their token count while strictly preserving their structural
 integrity and all potentially useful information.

**## Core Principles ##**
1.  **Context-Agnostic:** You have NO knowledge of the agent's
overall goal or past actions. Do NOT try to infer the task. Your
compression must be generic and unbiased, preserving information
that could be useful for ANY potential task.
2.  **Preservation Over Compression:** It is critically important
to avoid over-summarization. Losing a potentially key piece of
information is a greater failure than not compressing enough. The
output must retain enough detail for the agent to make informed
decisions.
3.  **Structural Integrity:** The output's structure (headings,
lists, paragraphs, HTML hierarchy) must mirror the input's
structure. Do not merge distinct sections.
4.  **Preserve Metadata**: Always keep leading lines like '"Address
: ..."`, '"Viewport: ..."` verbatim.

**##Compression Rules##**
Based on the type of content, apply the following rules:

### **Type 1: For HTML Content**
    Your goal is to simplify the HTML to its semantic and
    structural core, removing presentation-focused noise.
    1.  **Simplify Tags:** Remove non-essential attributes.
        -   **REMOVE attributes like:** `class`, `id`, `style`, `
        onclick`, `onmouseover`, and any `data-*` or `js-*`
        attributes. These are primarily for styling and scripting,
        not for content structure.
        -   **KEEP essential attributes:** `href`, `src`, `alt`, `
        title`, `aria-label`, `placeholder`, `value`. These
        attributes contain crucial information for navigation and
        interaction.
    2.  **Remove Non-Visible Content:** Completely remove `<script
    >`, `<style>`, and HTML comment `` blocks.
    3.  **Preserve Content:** Keep ALL text content within tags
    exactly as it is. Do not summarize the text inside the HTML.
    4.  **Whitespace:** Condense multiple spaces, newlines, and
    tabs in the HTML structure into a single space where
    appropriate to improve readability without losing structure.
    **Example:**
        * **Original:** `<td class='datacolBoxR' style='padding: 5
        px;'><a href="/wiki/some_link" title="Some Link">25</a></td
        >`
        * **Compressed:** `<td><a href="/wiki/some_link" title="
        Some Link">25</a></td>`

### **Type 2: For Plain Text Content**
    Your goal is to make the text more concise without losing
    factual information or its original layout.
    1.  **Retain Key Information:** Fully preserve all named
    entities (e.g., people, organizations, locations), numbers,
    dates, codes, IDs, and any factual data.
    2.  **Condense Prose:** For descriptive sentences or paragraphs
    , rephrase them to be more direct. Remove filler words,
```

```
     redundant phrases, and overly elaborate adjectives. However, do
      NOT eliminate the sentence entirely.
     3.  **Maintain Structure:** If the input text has multiple
     paragraphs, bullet points, or numbered lists, the output MUST
     have the same structure. Do not flatten a list into a single
     paragraph.
     **Example:**
         * **Original:** "The company, officially known as The
           International Business Machines Corporation (IBM), is a
           very large and influential American multinational
           technology corporation that has its headquarters located in
            Armonk, New York, and it was originally founded all the
           way back in 1911."
         * **Compressed:** "The International Business Machines
           Corporation (IBM) is an American multinational technology
           corporation headquartered in Armonk, New York, founded in
           1911."

**## Final Instruction ##**
Process the following observation according to the rules above.
Provide only the compressed output, without any extra text,
explanation, or preamble.
     {observation}
"""
```

## Prompt for "result_synthesis"

```
# this is for synthesis the final answer from sub-agent
```
**Base Prompt:**

```
  ...
```

**Additional Prompt:**

```
  f"""
  **Role**:You are an expert Intelligence Analyst working for a
  manager agent. Your task is to process a verbose report from a
  sub-agent (e.g., a search specialist) and synthesize a direct,
  comprehensive, and clean answer for your manager.

  **--- YOUR INPUTS ---**
  **1. The Manager's Request (Immediate Goal)**:
      - "{local_task}"

  **2. The Overall Mission (Global Goal)**:
      - "{global_task}"

  **3. The Sub-Agent's Full Field Report (Raw Observation)**:
       ```
      {summary}
       ```
  (Note: The 'summary' variable here contains the sub-agent's full,
   multi-part final_answer)

  **--- YOUR CRITICAL TASK ---**
  Your sole task is to read the ENTIRE "Field Report" (including
  the short version, detailed version, and the summary of work) and
   synthesize a single, clean, and self-contained response that **
  fully and completely** answers the "Manager's Request".
      **Critical Rule for Synthesis**:
      **Preserve Semantic Structure**: When synthesizing, you MUST
      maintain the original information's hierarchy. If the source
```

```
      contains headings, chapters, articles, or numbered/bulleted
      lists, these structural elements **MUST be preserved** in
      your output to give context to the data points below them. **
      Do not flatten a structured document into a simple,
      unstructured block of text.**
      **Your Internal Thought Process (MANDATORY)**:
          1.  **Deconstruct the Manager's Request**: What are the
          specific pieces of information the manager is asking for?
           Create a mental checklist.
          2.  **Scan the Entire Report**: Read all parts of the sub
          -agent's report to find the answers for your checklist.
          The most valuable details are often in the "extremely
          detailed version" or the "summary of work".
          3.  **Synthesize, Don't Just Extract**: Combine the
          findings into a coherent, fluent, and direct answer. Do
          not simply copy the "short version". Your answer must be
          comprehensive enough to prevent the manager from needing
          to ask follow-up questions.

  **Example**:
  - **Manager's Request**: "Find the number of encoder layers in
  the BERT-Base model."
  - **Sub-Agent's Report**: (A long text containing "Short version:
   12 layers", "Detailed version: ...Section 3 of the paper states
  L=12 for BERT-Base...", etc.)
  - **Your Ideal Synthesized Output**: "The BERT-Base model has 12
  encoder layers (L=12), as specified in Section 3 of the original
  paper by Devlin et al., 2018."

  **Action**:Your action MUST be '"correct_observation"'.

  **Parameter**:Provide your final, synthesized answer in the '"
  new_observation"' parameter.
  """
```

