# OpenReview forum: "Stop Wasting Your Tokens: Towards Efficient Runtime Multi-Agent Systems"
_ICLR.cc/2026/Conference — ICLR 2026 Poster_

### Official Review · Reviewer_WiAV · 2025-10-22

**Soundness:** 3
**Presentation:** 2
**Contribution:** 3
**Rating:** 4
**Confidence:** 5

**Summary:**

The paper introduces SUPERVISORAGENT, a lightweight, runtime “supervisor” that sits on top of a multi-agent system (MAS). A non-LLM prioritized adaptive filter detects four trigger conditions (sub-agent completion, error occurrence, inefficient behavior, excessive observation length) and, when activated, a three-stage intervention pipeline decides among four actions: approve, provide_guidance, correct_observation, run_verification. The goal is to reduce token usage and improve robustness without modifying underlying agents. Experiments on GAIA (Smolagent) and several code/math/QA benchmarks report sizable token savings with roughly maintained—or sometimes improved—accuracy, and an ablation suggests observation “purification” is the main driver of efficiency, while guidance/correction help accuracy.

**Strengths:**

1. The paper addresses a critical and highly relevant challenge in the field of agentic AI. The prohibitive token cost and lack of operational robustness are major barriers to the real-world deployment of complex Multi-Agent Systems. The work's focus on runtime efficiency is a valuable contribution.

2. The empirical validation is extensive and rigorous. The authors test their method on six different benchmarks spanning multiple domains, which strongly supports the claim of general applicability.

3. The paper highlights reduced variance and a more compact per-problem token distribution after supervision—an under-discussed but practically important property for cost predictability.

4. The prioritized LLM-free filter (sub-agent summary → error → inefficiency → length) is a neat engineering idea to avoid constant, costly oversight while still catching the most harmful failure modes. The concrete heuristics (e.g., length threshold ≈3,000 chars; loop/inefficiency checks) are spelled out sufficiently to reproduce.

**Weaknesses:**

1. The core idea—a meta-controller that detects errors/loops/long observations and either prunes context or issues guidance—resembles prior runtime oversight, routing, or budgeted-reasoning controllers. The paper would benefit from a sharper contrast to existing runtime monitors/filters (as opposed to offline compression or architectural agent redesign). As written, the conceptual delta may feel incremental.

2. The authors justify Smolagent because it relies less on “powerful external tools,” isolating the MAS. But in many real MAS workloads, tool calls dominate; it is unclear whether the same gains hold with tool-heavy agents, where tool output is the main token pressure and mis-tooling is the dominant failure mode. The paper acknowledges tokens meticulously but not end-to-end cost/latency including tool latencies and external compute.

3. The authors justify using Smolagent as the base framework because its capabilities stem from "internal agentic interactions rather than powerful external tools". While this provides a controlled environment, it also means the baseline might be inherently less efficient and robust. Applying SUPERVISORAGENT to a weaker baseline could inflate its perceived benefits.

4. The prioritized conditions (e.g., 3,000-char cutoff; loop detectors) appear hand-tuned. How sensitive are results to these thresholds across tasks and models? The paper claims model-agnosticism but does not deeply analyze failure cases where the supervisor fires too often/too rarely on different backbones.

**Questions:**

1. What fraction of ActionSteps trigger each condition on GAIA (by level) and on each external benchmark? Please report a table of invocations per action and supervisor token cost so readers can reason about net savings.

2. Why was the ablation study limited to the 30 most token-intensive tasks from the GAIA validation set? Do the individual contributions of the Correction, Guidance, and Purification modules remain consistent when evaluated across the entire dataset, including less token-intensive tasks?

3. The current filter is rule-based. Have you considered a learning-based approach for the supervisor? For example, could a policy be trained to decide when and how to intervene, potentially leading to a more adaptive and less heuristic-driven system?

4. Have you tested a tool-rich agent configuration (e.g., retrieval-augmented browsing with multi-source aggregation)? If the majority of cost/latency comes from tools rather than LLM tokens, does the supervisor still help?

---

> ### Author Response · Authors · 2025-11-23
> **Response to WiAV (1/n)**
>
> We sincerely appreciate the reviewer's rigorous review. We value the recognition of our work's runtime efficiency focus and the acknowledgment of the "neat engineering idea" behind our adaptive filter. The critiques regarding conceptual novelty and tool interactions have pushed us to significantly clarify our positioning and expand our empirical validation. Major updates in the revised PDF are marked in **blue**.
>
> ### **To W1: Conceptual Novelty and Runtime Process Control**.
>
> > The core idea—a meta-controller that detects errors/loops/long observations and either prunes context or issues guidance—resembles prior runtime oversight, routing, or budgeted-reasoning controllers. The paper would benefit from a sharper contrast to existing runtime monitors/filters (as opposed to offline compression or architectural agent redesign). As written, the conceptual delta may feel incremental.
>
> We clarify the fundamental conceptual delta between SupervisorAgent and prior runtime oversight methods, highlighting our distinct contribution to runtime efficiency.
>
> **Novel Paradigm: Efficiency-Centric Runtime Control**. While meta-level control exist for planning or accuracy (e.g., APO[1], MegaAgent[2]), SupervisorAgent is the first to repurpose this paradigm specifically for **Token Efficiency**. Unlike prior design-time optimizations[3] (e.g., pruning[4]) or static context compression[5], we introduce **runtime process control**. This fills a critical gap by **dynamically correcting inefficiencies as they occur**, rather than relying on static topologies.
>
> **Solving the "Watcher's Dilemma" via SMAS**. Standard monitors often negate efficiency gains through continuous, costly LLM queries. Our **LLM-free Adaptive Filter** solves this by ensuring zero-overhead monitoring until a specific trigger justifies intervention. This mechanism transforms the Supervisor from a mere router into a **conditional interventionist**. Consequently, we view SupervisorAgent not just as a specific agent, but as a **foundational component** for future scalable systems—a paradigm we define as **Supervised MAS (SMAS)**.
>
> **Distinction from Concurrent Work**. One of the comparable concurrent works is the **Guard Agent** in AWorld[6] (Aug 2025). However, as we clarify in the **Discussion**, the Guard focuses narrowly on factual verification for **accuracy**. In contrast, SupervisorAgent targets **system-wide cost control**. Our results (Table 4) show that our efficiency-focused supervision complements such accuracy-focused guards, yielding **superior net savings** (~36.54%).
>
> **Reference**
>
> [1] Li A, Xie Y, Li S, et al. Agent-oriented planning in multi-agent systems. ICLR 2025.
>
> [2] Wang Q, Wang T, Tang Z, et al. MegaAgent: A large-scale autonomous LLM-based multi-agent system without predefined SOPs. ACL Findings 2025.
>
> [3] Zhang G, Niu L, Fang J, et al. Multi-agent architecture search via agentic supernet. ICML 2025.
>
> [4] Wang Z, Wang Y, Liu X, et al. AgentDropout: Dynamic agent elimination for token-efficient and high-performance LLM-based multi-agent collaboration. ACL 2025.
>
> [5] Chen J, Liang J, Wang B. Smurfs: Multi-agent system using context-efficient DFSDT for tool planning. NAACL 2025.
>
> [6] Xie Z, Wu Q, Yu C, et al. Profile-Aware Maneuvering: A dynamic multi-agent system for robust GAIA problem solving by AWorld. arXiv:2508.09889, 2025.

---

> > ### Author Response · Authors · 2025-11-23
> > **Response to WiAV (2/n)**
> >
> > ### **To W2, W3, & Q4: Generalization to Tool-Heavy Frameworks**.
> >
> > > The authors justify Smolagent because it relies less on “powerful external tools,” isolating the MAS. But in many real MAS workloads, tool calls dominate; it is unclear whether the same gains hold with tool-heavy agents, where tool output is the main token pressure and mis-tooling is the dominant failure mode. The paper acknowledges tokens meticulously but not end-to-end cost/latency including tool latencies and external compute.
> >
> > > The authors justify using Smolagent as the base framework because its capabilities stem from "internal agentic interactions rather than powerful external tools". While this provides a controlled environment, it also means the baseline might be inherently less efficient and robust. Applying SUPERVISORAGENT to a weaker baseline could inflate its perceived benefits.
> >
> > > Have you tested a tool-rich agent configuration (e.g., retrieval-augmented browsing with multi-source aggregation)? If the majority of cost/latency comes from tools rather than LLM tokens, does the supervisor still help?
> >
> > We agree that validating performance on tool-rich agents is critical to prove the framework's robustness beyond "internal" reasoning.
> >
> > **Validation on Strong, Tool-Heavy Baselines**. To address the concern about "inflated benefits on weak baselines," we integrated SupervisorAgent into **OAgents** and **AWorld** (detailed in Appendix A.4.2).
> >
> > - **Tool-Centric Baseline**: **OAgents** is engineered for multi-source browsing with extensive tool libraries, while **AWorld** employs a hierarchical architecture with frequent external interactions via MCP.
> >
> > - **Empirical Results**: As shown in **Table 4**, SMAS is highly effective even on these robust and tool-intensive baselines, reducing token costs by **39.36% (OAgents)** and **36.54% (AWorld)**. This empirically proves that our gains are not artifacts of a weak baseline but are **robust improvements for tool-heavy workflows**.
> >
> > **Table: Cross-framework performance of SupervisorAgent on GAIA subset** (top-10 most token-intensive tasks per level).
> >
> > | Method                       | Avg. Acc.     | Avg. Token   | L1 Avg. Token | L2 Avg. Token | L3 Avg. Token |
> > |------------------------------|---------------|--------------|--------------------|--------------------|--------------------|
> > | Smolagent                    | 40.00         | 1,446,526    | 933,013            | 2,037,437          | 1,369,131          |
> > | **+ SMAS**                   | 46.67 ↑6.67%  | 721,332 ↓50.13% | 522,364 ↓44.01%   | 960,694 ↓52.85%    | 680,939 ↓50.26%    |
> > | AWorld (without Guard)       | 23.33         | 155,239      | 50,851             | 217,332            | 166,500            |
> > | AWorld (with Guard)          | 30.00         | 353,738      | 135,413            | 463,083            | 376,878            |
> > | **AWorld (with SMAS)**       | 36.67 ↑6.67%  | 224,480 ↓36.54% | 90,569 ↓33.12%    | 355,051 ↓23.33%    | 194,561 ↓48.38%    |
> > | OAgents                      | 46.67         | 530,939      | 430,852            | 359,511            | 802,454            |
> > | **+ SMAS**                   | 46.67         | 321,957 ↓39.36% | 214,604 ↓50.19%   | 274,875 ↓23.54%    | 476,393 ↓40.63%    |
> >
> > **Addressing Tool Latency & External Cost**. The reviewer correctly note that "mis-tooling" and tool latency are dominant factors. While we defer a standardized metric for external API pricing to future work, we argue that SupervisorAgent **directly mitigates** these external costs through **two mechanisms**:
> >
> > - **Reduction of Tool Calls**: By breaking inefficient loops, the Supervisor reduces the frequency of external invocations. For instance, in our **case study** (Figure 2d, Appendix A.6), SMAS reduced the total steps **from 23 to 13** (a **43%** reduction), directly eliminating **70%** token cost. Fewer calls translate immediately to lower external compute costs and latency.
> >
> > - **Prevention of Mis-tooling**: By purifying noisy tool outputs, we prevent the agent from **overlooking valid information** buried in the noise. This is evidenced by the **36.54%** token savings on **AWorld**. Without purification, agents often fail to spot the retrieved answer within the excessive context, leading them to mistakenly re-invoke the same tools to fetch data they actually already possess. SupervisorAgent eliminates these redundant "blind" calls, ensuring the agent correctly identifies that a sub-goal is complete and moves to the next logical step.

---

> > > ### Author Response · Authors · 2025-11-23
> > > **Response to WiAV (3/n)**
> > >
> > > ### **To W4: Heuristics and Hyperparameter Sensitivity**.
> > >
> > > > The prioritized conditions (e.g., 3,000-char cutoff; loop detectors) appear hand-tuned. How sensitive are results to these thresholds across tasks and models? The paper claims model-agnosticism but does not deeply analyze failure cases where the supervisor fires too often/too rarely on different backbones.
> > >
> > > We address the concerns regarding hyperparameter subjectivity and backbone-dependent behaviors from two aspects:
> > >
> > > 1. **Robustness of Heuristic Thresholds**. We focused our sensitivity analysis on the observation threshold ($\tau_{\text{len}}$) as it is the **dominant factor** influencing token efficiency (purification).
> > >
> > > - **Sensitivity Analysis**: As detailed in **Appendix A.3.2** and **Figure 5**, testing $\tau_{\text{len}}$ from **1,000 to 7,000** confirms the system's **robustness**. Performance remains stable without significant degradation, mitigating concerns about brittleness.
> > >
> > > - **Adaptability (Not just "Hand-tuned")**: We emphasize that these parameters are **adaptable structural constraints** rather than rigid numbers. For instance, we adjusted $\tau_{\text{len}}$ to 10,000 for the more verbose **OAgents** framework, resulting in Token Reduction of **39.36%** on GAIA subset. This confirms the parameter **logically correlates with agent architecture** rather than being arbitrarily overfitted.
> > >
> > > 2. **Analysis of Firing Frequency across Backbones**. The reviewer rightly pointed out the need to analyze behavior across backbones. We have included a dedicated **Failure Mode Analysis (Appendix A.5)** to address when the supervisor fires too often or rarely. A short summary is provided below:
> > >
> > > - **"Firing Too Often" (Weaker Models)**: Our logs with less capable models (e.g., Qwen3-235B) show they trigger the Error Correction module significantly more often due to basic failures (e.g., malformed JSON). This increases overhead and reduces net savings compared to GPT-4.1.
> > >
> > > - **"Firing Too Rarely" (Stronger Models)**: Conversely, stronger models may occasionally bypass supervision if the heuristic filter is not sensitive enough to subtle logic errors that do not manifest as loops or overflows.
> > >
> > > This analysis clarifies that while the framework is model-agnostic, the **efficiency gains** naturally correlate with the **backbone's capability**.
> > >
> > > ### **To Q3: Learning-Based Filter**.
> > >
> > > > The current filter is rule-based. Have you considered a learning-based approach for the supervisor? For example, could a policy be trained to decide when and how to intervene, potentially leading to a more adaptive and less heuristic-driven system?
> > >
> > > This is an excellent suggestion. We agree that a learning-based policy could offer finer granularity than heuristics. However, our current heuristic design was a deliberate choice to prioritize **zero-shot, plug-and-play deployability**.
> > >
> > > - A learning-based filter would require collecting interaction trajectories and training a policy for each specific MAS architecture, **increasing the barrier to entry**. In contrast, our heuristic filter **works out-of-the-box** across diverse frameworks (Smolagent, AWorld, OAgents) without requiring warm-up or training data.
> > >
> > > - Nevertheless, evolving this into a "Self-Evolving Supervision Policy" is the logical next step to break the ceiling of heuristic rules. We have explicitly added this to our **Future Directions (line 525)** as a promising avenue for research.

---

> > > > ### Author Response · Authors · 2025-11-23
> > > > **Response to WiAV (4/n)**
> > > >
> > > > ### **To Q1: Overhead, Net Savings, and Trigger Frequency**.
> > > >
> > > > > What fraction of ActionSteps trigger each condition on GAIA (by level) and on each external benchmark? Please report a table of invocations per action and supervisor token cost so readers can reason about net savings.
> > > >
> > > > We provide a granular analysis of the supervisor's intervention frequency and a transparent breakdown of the net token savings to clarify the cost-benefit dynamics.
> > > >
> > > > **Trigger Frequency Analysis**. Regarding the fraction of steps triggering supervision, our updated ablation study (Table 3) reveals the relative frequency and impact of each module:
> > > >
> > > > - **High Frequency**: *Observation Purification* is the most frequently triggered condition. Its removal causes the most **drastic drop** in token savings (from 29.68% to 15.96%), indicating it is the **primary source** of both Supervisor overhead and system-wide savings.
> > > >
> > > > - **Low Frequency**: In contrast, *Error Correction and Inefficiency Guidance* trigger significantly less often. Their primary contribution is **maintaining success rates** (Robustness) rather than bulk token reduction.
> > > >
> > > > **Net Savings Breakdown**. We confirm that all reported results are "**Net Savings**," fully accounting for the Supervisor's overhead. To enable precise reasoning about these savings, we provide a detailed cost breakdown in **Table 6 and Figure 6** (detailed in lines 480 & 1133).
> > > >
> > > > - We report Token Cost rather than raw Invocation Counts because **the cost per invocation varies drastically** (e.g., Purifyinga 200k-character observation costs orders of magnitude more than a simple Guidance prompt). A simple count table would be misleading for calculating net savings.
> > > >
> > > > - As shown in the table below, on GAIA pass@1, the Supervisor incurs an overhead (the difference between SMAS and SMAS (NET)). However, this investment reduces the Base MAS's consumption by **40.49%**, resulting in a solid **Net Reduction** of **29.68%**.
> > > >
> > > > **Table: Token efficiency analysis on GAIA validation set.** Comparison of token consumption across different pass@k settings.
> > > >
> > > > | Method                              | Avg. Tokens (K)            | L1 Tokens (K)              | L2 Tokens (K)              | L3 Tokens (K)              |
> > > > |-------------------------------------|----------------------------|----------------------------|----------------------------|----------------------------|
> > > > | **pass@1**                          |                            |                            |                            |                            |
> > > > | Smolagent                           | 527.76                     | 298.51                     | 619.59                     | 691.33                     |
> > > > | + SMAS                              | 314.07 ↓40.49%             | 220.63 ↓26.09%             | 342.18 ↓44.77%             | 411.58 ↓40.47%             |
> > > > | + SMAS (NET)                        | 371.12 ↓29.68%             | 258.28 ↓13.48%             | 404.96 ↓34.64%             | 489.22 ↓29.23%             |
> > > > | **pass@2**                          |                            |                            |                            |                            |
> > > > | Smolagent                           | 467.19                     | 275.85                     | 548.02                     | 589.92                     |
> > > > | + SMAS                              | 329.51 ↓29.47%             | 231.96 ↓15.91%             | 354.21 ↓35.37%             | 446.64 ↓24.29%             |
> > > > | + SMAS (NET)                        | 389.55 ↓16.62%             | 270.07 ↓2.10%              | 420.97 ↓23.18%             | 529.20 ↓10.29%             |
> > > > | **pass@3**                          |                            |                            |                            |                            |
> > > > | Smolagent                           | 502.40                     | 282.14                     | 605.05                     | 611.87                     |
> > > > | + SMAS                              | 312.06 ↓37.89%             | 236.28 ↓16.25%             | 342.36 ↓43.42%             | 366.31 ↓40.13%             |
> > > > | + SMAS (NET)                        | 369.52 ↓26.45%             | 276.84 ↓1.88%              | 409.05 ↓32.39%             | 427.72 ↓30.10%             |

---

> > > > > ### Author Response · Authors · 2025-11-23
> > > > > **Response to WiAV (5/n)**
> > > > >
> > > > > ### **To Q2: Consistency of Token Reduction Rates**.
> > > > >
> > > > > > Why was the ablation study limited to the 30 most token-intensive tasks from the GAIA validation set? Do the individual contributions of the Correction, Guidance, and Purification modules remain consistent when evaluated across the entire dataset, including less token-intensive tasks?
> > > > >
> > > > > We have expanded our ablation study to the **full** GAIA validation set, confirming that the modular contributions remain consistent across the entire task distribution.
> > > > >
> > > > > **Consistency Confirmed via Full-Set Analysis**. To eliminate ambiguity, we have **replaced** the subset ablation with a comprehensive study on the **full GAIA validation set** (see **Table 3**, line 427).
> > > > >
> > > > > - As shown in the table below, **the individual contributions remain consistent** with our initial findings: **Purification** is the dominant driver of **efficiency** (disabling it causes savings to drop from 29.68% to 15.96%), while **Correction and Guidance** are critical for maintaining **accuracy**.
> > > > >
> > > > > - We retained the original **subset analysis** in **Appendix A.4.2**, as it provides valuable insight into how the Supervisor's utility scales in **high-complexity regimes** where context management is most critical.
> > > > >
> > > > > **Table: Ablation study of SupervisorAgent's components** on the full GAIA validation set.
> > > > >
> > > > > | Method                       | Avg. Acc. | Avg. Token     | L1 Avg. Token | L2 Avg. Token | L3 Avg. Token |
> > > > > |------------------------------|-----------|----------------|--------------------|--------------------|--------------------|
> > > > > | Smolagent                    | 50.91     | 527,759        | 298,506            | 619,591            | 691,331            |
> > > > > | + SMAS (w/o Correction)      | 47.88     | 354,226 ↓32.88%| 221,515 ↓25.79%    | 363,871 ↓41.27%    | 592,852 ↓14.24%    |
> > > > > | + SMAS (w/o Guidance)        | 48.48     | 363,644 ↓31.10%| 253,591 ↓15.05%    | 419,913 ↓32.23%    | 401,861 ↓41.87%    |
> > > > > | + SMAS (w/o Purification)    | 49.70     | 443,520 ↓15.96%| 270,058 ↓9.53%     | 502,937 ↓18.83%    | 600,582 ↓13.13%    |
> > > > > | + SMAS                       | 50.91     | 371,119 ↓29.68%| 258,279 ↓13.48%    | 404,955 ↓34.64%    | 489,222 ↓29.23%    |

---

> ### Author Response · Authors · 2025-11-28
> **Response to WiAV (6/n)**
>
> Dear reviewer WiAV,
>
> We truly appreciate the time and effort you’ve dedicated to reviewing our submission. We’ve replied to your questions and made revisions, particularly addressing methodology concerns highlighted as a weakness. Additionally, we’ve incorporated feedback from other reviewers, which may also help clarify any additional questions you might have.
>
> Since the discussion phase ends soon, we wanted to follow up and would greatly value any further thoughts or concerns you might have so we can address them appropriately.
>
> Thank you again for your time and commitment to the review process.

---

### Official Review · Reviewer_UCU6 · 2025-10-25

**Soundness:** 3
**Presentation:** 3
**Contribution:** 3
**Rating:** 6
**Confidence:** 3

**Summary:**

This paper addresses critical challenges in MAS, such as excessive token consumption and error propagation, by introducing SUPERVISORAGENT, a lightweight, non-intrusive meta-agent framework for runtime supervision. The approach is innovative in its use of an LLM-free adaptive filter to trigger targeted interventions, enabling proactive error correction, inefficiency guidance, and observation purification without modifying the base agents' architecture. The experiments are comprehensive, spanning multiple benchmarks and models, and demonstrate efficiency gains on a specific baseline. However, the paper's claims of a "significant Pareto improvement" are undermined by unfavorable comparisons to SOTA baselines presented in its own results, raising concerns about the method's broader applicability and impact.

**Strengths:**

The paper effectively highlights underexplored pain points in MAS, such as runtime inefficiencies from error propagation and excessive observations, which lead to high token costs (for example, up to 2 million tokens on GAIA tasks). This framing is fresh and relevant, building on recent MAS advancements while addressing a critical paradox: increased autonomy often reduces robustness and economic viability.

SUPERVISORAGENT is a lightweight, non-intrusive framework that integrates seamlessly with existing MAS without architectural changes. Its LLM-free adaptive filter for triggering interventions (e.g., at agent-agent, agent-tool, or agent-memory points) is efficient and proactive, focusing on runtime process control rather than static optimizations. The action space (e.g., approve, provide_guidance, correct_observation, run_verification) is well-formalized, and the memory-augmented context window enables holistic oversight. This design is modular and broadly applicable, as validated across diverse benchmarks and models (e.g., GPT-4.1, Gemini-2.5-pro, Qwen3).

**Weaknesses:**

The paper claims a "significant Pareto improvement" (reduced tokens without compromising success rates), but this holds only against the weak Smolagent baseline (50.91% accuracy, 527.76K tokens reduced to 371.12K). In contrast, the paper's own Table 1 shows SOTA baseline AWorld achieving higher accuracy (60.00%) at much lower cost (128.27K tokens). The supervised system (Smolagent + SMAS) is thus both less accurate and more expensive than AWorld, contradicting the Pareto claim. While the authors justify Smolagent for isolating "agentic interactions" over external tools, this does not excuse avoiding direct SOTA integration or comparison. It makes the method's impact limited.

The LLM-free filter relies on hardcoded heuristics (e.g., detecting repetitive actions or observations exceeding 3,000 characters). These thresholds lack justification, sensitivity analysis, or ablation, raising concerns about brittleness and potential overfitting to Smolagent or GAIA. Without robustness tests (e.g., varying thresholds across datasets), the filter's generalizability is questionable.

Token reductions vary inexplicably between main results (29.68% on full GAIA) and ablation on 30 high-token tasks (50.13%). This suggests benefits are skewed toward long-tail cases, potentially masking uneven performance. The abstract and conclusions claim "average" savings (e.g., 29.45%) without clarifying this distribution, which could mislead readers and requires more transparent analysis.

**Questions:**

Why was SUPERVISORAGENT not applied to SOTA baselines like AWorld? If integrated, could it further reduce AWorld's token costs (e.g., from 128K to below 100K) while maintaining its 60% accuracy? If incompatible, please explicitly discuss this limitation in the paper.

How does SUPERVISORAGENT's own overhead (e.g., from context aggregation or LLM calls in interventions) factor into total token costs?

---

> ### Author Response · Authors · 2025-11-23
> **Response to Reviewer UCU6 (1/n)**
>
> We sincerely thank the reviewer for the comprehensive review. We value the reviewer's recognition of our work as a "fresh and relevant" approach to the autonomy-efficiency paradox. The reviewer's rigorous scrutiny regarding SOTA comparisons and methodological robustness has pushed us to substantially strengthen our empirical validation. In the revised PDF, major updates are marked in **blue**.
>
> ### **To W1 & Q1: Pareto Improvement and SOTA Comparison**.
>
> > The paper claims a "significant Pareto improvement" (reduced tokens without compromising success rates), but this holds only against the weak Smolagent baseline (50.91% accuracy, 527.76K tokens reduced to 371.12K). In contrast, the paper's own Table 1 shows SOTA baseline AWorld achieving higher accuracy (60.00%) at much lower cost (128.27K tokens). The supervised system (Smolagent + SMAS) is thus both less accurate and more expensive than AWorld, contradicting the Pareto claim. While the authors justify Smolagent for isolating "agentic interactions" over external tools, this does not excuse avoiding direct SOTA integration or comparison. It makes the method's impact limited.
>
> > Why was SUPERVISORAGENT not applied to SOTA baselines like AWorld? If integrated, could it further reduce AWorld's token costs (e.g., from 128K to below 100K) while maintaining its 60% accuracy? If incompatible, please explicitly discuss this limitation in the paper.
>
> We appreciate this critical observation. We clarify that our initial "Pareto improvement" claim referred to the **self-enhancement** of a base system (i.e., Smolagent + SMAS vs. Smolagent), rather than asserting immediate superiority over all external architectures. We acknowledge that comparing a lightweight framework like Smolagent directly against a highly engineered SOTA like AWorld was structurally unequal.
>
> **Integrating with SOTA (AWorld & OAgents)**. We integrated our framework into **AWorld** and **OAgents** on GAIA subset (detailed in line 465, **Table 4**). The results answer the reviewer's question: *"Could it further reduce AWorld's token costs?"* **Yes.** Even on these highly optimized, tool-heavy systems, SMAS achieved significant reductions. It reduces token costs by **36.54%** (AWorld) and **39.36%** (OAgents) while maintaining or improving accuracy.
>
> **Table: Cross-framework performance of SupervisorAgent on GAIA subset** (top-10 most token-intensive tasks per level).
>
> | Method                       | Avg. Acc.     | Avg. Token   | L1 Avg. Token | L2 Avg. Token | L3 Avg. Token |
> |------------------------------|---------------|--------------|--------------------|--------------------|--------------------|
> | Smolagent                    | 40.00         | 1,446,526    | 933,013            | 2,037,437          | 1,369,131          |
> | **+ SMAS**                   | 46.67 ↑6.67%  | 721,332 ↓50.13% | 522,364 ↓44.01%   | 960,694 ↓52.85%    | 680,939 ↓50.26%    |
> | AWorld (without Guard)       | 23.33         | 155,239      | 50,851             | 217,332            | 166,500            |
> | AWorld (with Guard)          | 30.00         | 353,738      | 135,413            | 463,083            | 376,878            |
> | **AWorld (with SMAS)**       | 36.67 ↑6.67%  | 224,480 ↓36.54% | 90,569 ↓33.12%    | 355,051 ↓23.33%    | 194,561 ↓48.38%    |
> | OAgents                      | 46.67         | 530,939      | 430,852            | 359,511            | 802,454            |
> | **+ SMAS**                   | 46.67         | 321,957 ↓39.36% | 214,604 ↓50.19%   | 274,875 ↓23.54%    | 476,393 ↓40.63%    |
>
> This confirms that SupervisorAgent is not merely a patch for "weak" baselines (like Smolagent) but a **Universal Efficiency Layer** (or Supervised MAS, **SMAS**) that can be plugged into future designs to further optimize their operational costs and robustness.

---

> > ### Author Response · Authors · 2025-11-23
> > **Response to Reviewer UCU6 (2/n)**
> >
> > ### **To W2: Robustness of Heuristic Filters**.
> >
> > > The LLM-free filter relies on hardcoded heuristics (e.g., detecting repetitive actions or observations exceeding 3,000 characters). These thresholds lack justification, sensitivity analysis, or ablation, raising concerns about brittleness and potential overfitting to Smolagent or GAIA. Without robustness tests (e.g., varying thresholds across datasets), the filter's generalizability is questionable.
> >
> > We addressed the concern regarding the "hardcoded" nature of the filter by expanding our analysis to demonstrate both **robustness and adaptability**.
> >
> > **Sensitivity Analysis**. As detailed in **Appendix A.3.2** and **Figure 5**, we conducted a sensitivity analysis on the observation threshold ($\tau_{\text{len}}$) ranging from **1,000 to 7,000** characters. The results confirm that performance (token savings and success rate) remains **stable** across this wide range, mitigating concerns about overfitting to a specific "3,000" number.
> >
> > **Adaptability to Architecture**. Crucially, we demonstrate that these thresholds are adaptable structural parameters. For the **OAgents** framework, which is inherently more verbose due to complex tool usage, we adjusted $\tau_{\text{len}}$ to **10,000** (line 1079). This successfully triggered the same efficiency gains, proving that the filter logic is **generalizable**—it scales logically with the agent's verbosity profile rather than being rigidly overfitted to Smolagent.
> >
> > ### **To W3: Consistency of Token Reduction Rates**.
> >
> > > Token reductions vary inexplicably between main results (29.68% on full GAIA) and ablation on 30 high-token tasks (50.13%). This suggests benefits are skewed toward long-tail cases, potentially masking uneven performance. The abstract and conclusions claim "average" savings (e.g., 29.45%) without clarifying this distribution, which could mislead readers and requires more transparent analysis.
> >
> > We have revised our analysis to ensure full transparency regarding the distribution of token savings and to address the concern about performance consistency.
> >
> > **Eliminating Ambiguity**. The discrepancy the reviewer noted (~29% vs. ~50%) arose because the initial ablation focused solely on "long-tail" high-cost tasks to verify **maximum impact**. We agree this could be misleading regarding average performance. We have **replaced** the ablation analysis in the main text with a new study on the **full GAIA validation set (Table 3, line 427)**.
> >
> > **Consistency Confirmed**. This comprehensive analysis confirms that the **mechanism remains consistent** across the general distribution: **Purification** is the dominant driver of **efficiency** (disabling it causes savings to drop from 29.68% to 15.96%), while **Correction and Guidance** are critical for maintaining **accuracy**.
> >
> > **Ablation study of SupervisorAgent's components** on the full GAIA validation set.
> >
> > | Method                       | Avg. Acc. | Avg. Token     | L1 Avg. Token | L2 Avg. Token | L3 Avg. Token |
> > |------------------------------|-----------|----------------|--------------------|--------------------|--------------------|
> > | Smolagent                    | 50.91     | 527,759        | 298,506            | 619,591            | 691,331            |
> > | + SMAS (w/o Correction)      | 47.88     | 354,226 ↓32.88%| 221,515 ↓25.79%    | 363,871 ↓41.27%    | 592,852 ↓14.24%    |
> > | + SMAS (w/o Guidance)        | 48.48     | 363,644 ↓31.10%| 253,591 ↓15.05%    | 419,913 ↓32.23%    | 401,861 ↓41.87%    |
> > | + SMAS (w/o Purification)    | 49.70     | 443,520 ↓15.96%| 270,058 ↓9.53%     | 502,937 ↓18.83%    | 600,582 ↓13.13%    |
> > | + SMAS                       | 50.91     | 371,119 ↓29.68%| 258,279 ↓13.48%    | 404,955 ↓34.64%    | 489,222 ↓29.23%    |
> >
> > We have moved the original **subset analysis** to **Appendix A.4.2**, as it provides valuable insight into how the Supervisor's utility scales up in extreme, high-cost scenarios where token savings are most critical.

---

> > > ### Author Response · Authors · 2025-11-23
> > > **Response to Reviewer UCU6 (3/n)**
> > >
> > > ### **To Q2: Overhead and Net Savings**.
> > >
> > > > How does SUPERVISORAGENT's own overhead (e.g., from context aggregation or LLM calls in interventions) factor into total token costs?
> > >
> > > We confirm that all reported token costs are "**Net Costs**," fully accounting for the **Supervisor's overhead**. To make this explicitly clear, we provided a breakdown in **Table 6 and Figure 6** (lines 480 & 1133). As shown below, the Supervisor itself incurs a cost (averaging **15%** of total usage). However, this investment yields a much larger reduction in the Base MAS's consumption (**40%** reduction).
> > >
> > > **Table: Token efficiency analysis on GAIA validation set.** Comparison of token consumption across different pass@k settings.
> > >
> > > | Method                              | Avg. Tokens (K)            | L1 Tokens (K)              | L2 Tokens (K)              | L3 Tokens (K)              |
> > > |-------------------------------------|----------------------------|----------------------------|----------------------------|----------------------------|
> > > | **pass@1**                          |                            |                            |                            |                            |
> > > | Smolagent                           | 527.76                     | 298.51                     | 619.59                     | 691.33                     |
> > > | + SMAS                              | 314.07 ↓40.49%             | 220.63 ↓26.09%             | 342.18 ↓44.77%             | 411.58 ↓40.47%             |
> > > | + SMAS (NET)                        | 371.12 ↓29.68%             | 258.28 ↓13.48%             | 404.96 ↓34.64%             | 489.22 ↓29.23%             |
> > > | **pass@2**                          |                            |                            |                            |                            |
> > > | Smolagent                           | 467.19                     | 275.85                     | 548.02                     | 589.92                     |
> > > | + SMAS                              | 329.51 ↓29.47%             | 231.96 ↓15.91%             | 354.21 ↓35.37%             | 446.64 ↓24.29%             |
> > > | + SMAS (NET)                        | 389.55 ↓16.62%             | 270.07 ↓2.10%              | 420.97 ↓23.18%             | 529.20 ↓10.29%             |
> > > | **pass@3**                          |                            |                            |                            |                            |
> > > | Smolagent                           | 502.40                     | 282.14                     | 605.05                     | 611.87                     |
> > > | + SMAS                              | 312.06 ↓37.89%             | 236.28 ↓16.25%             | 342.36 ↓43.42%             | 366.31 ↓40.13%             |
> > > | + SMAS (NET)                        | 369.52 ↓26.45%             | 276.84 ↓1.88%              | 409.05 ↓32.39%             | 427.72 ↓30.10%             |
> > >
> > > Consequently, the "Total System Cost" (Base MAS+ Supervisor) is significantly lower than the "Baseline Cost," validating the **net-positive** impact of our approach.

---

### Official Review · Reviewer_vopL · 2025-10-31

**Soundness:** 4
**Presentation:** 3
**Contribution:** 3
**Rating:** 6
**Confidence:** 3

**Summary:**

This paper addresses two critical and intertwined problems in modern Multi-Agent Systems (MAS): operational inefficiency (excessive token consumption) and lack of robustness (failures from error propagation). The authors argue that existing methods are either post-hoc (analyzing failures after they occur) or design-time (optimizing the agent architecture itself), with a lack of tools for proactive, real-time intervention .




To fill this gap, the authors propose SUPERVISORAGENT, a lightweight, modular meta-agent framework designed for runtime supervision. The framework's core is a lightweight, LLM-free adaptive filter that monitors agent interactions. This filter triggers interventions only at critical junctures, which it identifies as: 1) explicit error occurrences, 2) inefficient behaviors (like loops), or 3) excessive observation lengths.





When triggered, the SUPERVISORAGENT (which is an LLM) analyzes the system's state using a rich context window and performs one of four actions: approve, provide_guidance, correct_observation, or run_verification .


The authors demonstrate the system's effectiveness by applying it to the Smolagent framework on the GAIA benchmark, achieving a 29.45% reduction in token consumption without any loss in task success rate. The paper further validates this approach's generalizability across five other benchmarks (for math, code, and QA) and multiple state-of-the-art foundation models .

**Strengths:**

The paper's framing of MAS inefficiency as a runtime process control problem is a fresh and valuable perspective.


The hybrid "LLM-free filter + LLM supervisor"  is a very strong and practical design choice that balances cost and capability.


The 29.45% token reduction on GAIA with no accuracy loss is a headline-worthy result, strongly supported by the data.



The ablation in Table 3 is a highlight, perfectly justifying the three-part design of the supervisor's intervention strategies (Purification for efficiency, Correction/Guidance for robustness) .


The method is shown to be model-agnostic (Figure 4b) and benchmark-agnostic (Table 2), proving it is a general framework and not a one-off trick.

**Weaknesses:**

1. The paper's main claim is "Stop Wasting Your Tokens", and it reports a ~30% token reduction. However, it appears this 30% saving applies only to the base agents (Smolagent). The paper never states the **token cost of the SUPERVISORAGENT itself**. The supervisor is an LLM (e.g., GPT-4.1)  and is called every time the filter is triggered. To make a true claim of efficiency, the paper must report the net token savings (i.e., Baseline_Tokens - (SMAS_Agent_Tokens + Supervisor_Agent_Tokens)). Without this, the primary claim of token reduction is incomplete.

2. The paper focuses exclusively on token efficiency. However, the proposed framework introduces an extra LLM call into the execution loop every time an intervention occurs. This will "stop the world" and add wall-clock latency. It's possible the system saves tokens but becomes significantly slower to run. A comprehensive "efficiency" analysis should include latency, especially when the interventions are designed to cut down on inefficient loops (which also consume time).

**Questions:**

1. Do the token savings reported in Table 1 and 2 (e.g., the 29.68% on GAIA)  account for the tokens consumed by the SUPERVISORAGENT itself? If not, what is the net token savings for the entire system (base agents + supervisor)?

2. The ablation study (Table 3) shows that "Purification" is the main source of token savings, while "Correction" and "Guidance" are key for accuracy . This suggests that for tasks where the baseline is already robust (high accuracy), a "Purification-Only" supervisor might offer the best cost-benefit. Have you considered this variant?

3. The "Excessive Observation Length" trigger is a hardcoded "3,000 characters". How sensitive is the framework's performance (both token savings and accuracy) to this hyperparameter?

---

> ### Author Response · Authors · 2025-11-23
> **Response to Reviewer vopL (1/n)**
>
> We sincerely thank the reviewer for the encouraging review and for highlighting our work as a "fresh and valuable perspective" with "headline-worthy" results. The reviewer's constructive feedback regarding net savings and latency has significantly improved the rigor of our manuscript. In the revised PDF, major updates are marked in **blue**.
>
> ### **To W1 & Q1: Token Overhead and Net Savings**.
>
> > The paper's main claim is "Stop Wasting Your Tokens", and it reports a ~30% token reduction. However, it appears this 30% saving applies only to the base agents (Smolagent). The paper never states the token cost of the SUPERVISORAGENT itself. The supervisor is an LLM (e.g., GPT-4.1) and is called every time the filter is triggered. To make a true claim of efficiency, the paper must report the net token savings (i.e., Baseline Tokens - (SMAS Agent Tokens + Supervisor Agent Tokens)). Without this, the primary claim of token reduction is incomplete.
>
> > Do the token savings reported in Table 1 and 2 (e.g., the 29.68% on GAIA) account for the tokens consumed by the SUPERVISORAGENT itself? If not, what is the net token savings for the entire system (base agents + supervisor)?
>
> We confirm that all token savings reported in our initial submission (Table 1 & 2) are indeed "**Net Token Savings**", which fully account for the tokens consumed by the SupervisorAgent.
>
> To make this explicit, we have added a detailed breakdown in **Table 6** and **Figure 6** (detailed in lines 480 & 1133). As shown in the table below, the Supervisor itself incurs a modest overhead (averaging **~15%** of baseline usage). However, this investment yields a much larger reduction in the Base Agents' consumption (reducing it by **~40%**).
>
> Consequently, even after deducting the Supervisor's cost, the system achieves a **Net Token Reduction** of 29.68% (on GAIA pass@1). This substantiates our claim: **the efficiency gains significantly outweigh the operational overhead**.
>
> **Table: Token efficiency analysis on GAIA validation set.** Comparison of token consumption across different pass@k settings.
>
> | Method                              | Avg. Tokens (K)            | L1 Tokens (K)              | L2 Tokens (K)              | L3 Tokens (K)              |
> |-------------------------------------|----------------------------|----------------------------|----------------------------|----------------------------|
> | **pass@1**                          |                            |                            |                            |                            |
> | Smolagent                           | 527.76                     | 298.51                     | 619.59                     | 691.33                     |
> | + SMAS                              | 314.07 ↓40.49%             | 220.63 ↓26.09%             | 342.18 ↓44.77%             | 411.58 ↓40.47%             |
> | + SMAS (NET)                        | 371.12 ↓29.68%             | 258.28 ↓13.48%             | 404.96 ↓34.64%             | 489.22 ↓29.23%             |
> | **pass@2**                          |                            |                            |                            |                            |
> | Smolagent                           | 467.19                     | 275.85                     | 548.02                     | 589.92                     |
> | + SMAS                              | 329.51 ↓29.47%             | 231.96 ↓15.91%             | 354.21 ↓35.37%             | 446.64 ↓24.29%             |
> | + SMAS (NET)                        | 389.55 ↓16.62%             | 270.07 ↓2.10%              | 420.97 ↓23.18%             | 529.20 ↓10.29%             |
> | **pass@3**                          |                            |                            |                            |                            |
> | Smolagent                           | 502.40                     | 282.14                     | 605.05                     | 611.87                     |
> | + SMAS                              | 312.06 ↓37.89%             | 236.28 ↓16.25%             | 342.36 ↓43.42%             | 366.31 ↓40.13%             |
> | + SMAS (NET)                        | 369.52 ↓26.45%             | 276.84 ↓1.88%              | 409.05 ↓32.39%             | 427.72 ↓30.10%             |

---

> > ### Author Response · Authors · 2025-11-23
> > **Response to Reviewer vopL (2/n)**
> >
> > ### **To W2: Latency and Wall-Clock Time**.
> >
> > > The paper focuses exclusively on token efficiency. However, the proposed framework introduces an extra LLM call into the execution loop every time an intervention occurs. This will "stop the world" and add wall-clock latency. It's possible the system saves tokens but becomes significantly slower to run. A comprehensive "efficiency" analysis should include latency, especially when the interventions are designed to cut down on inefficient loops (which also consume time).
> >
> > We address the concern regarding the "stop-the-world" latency in **Table 7**. Crucially, our ablation study reveals that the architecture itself (the Adaptive Filter and the extra LLM calls for supervision) introduces **acceptable latency**:
> >
> > - **Architectural Efficiency**. The **SMAS (w/o Purification)** variant, which includes the filter and interventions for errors/loops, exhibits a runtime nearly identical to the baseline (**236.21s vs. 233.96s**, only **+0.96%**). This empirically proves that the "extra LLM call" mechanism is lightweight and does not significantly slow down the execution loop.
> >
> > - **Source of Latency**. The reported average latency increase (37.27%, **<1.5 mins/task**) is driven almost entirely by the **Purification** module, which processes massive observations.
> >
> > **Loop Reduction vs. Processing Time**. The reviewer correctly note that cutting inefficient loops should save time. Indeed, SMAS significantly reduces the **number of steps** (e.g., **43% reduction** in the case study, Figure 2d, Appendix A.6). However, the wall-clock time saved by skipping steps is currently outweighed by the compute-intensive nature of Purifying the remaining long contexts.
> >
> > We acknowledge this latency (<1.5 mins/task) but consider it a justifiable trade-off: we exchange **compute time** for **token savings** (~30%) and **robustness**, effectively converting "wasted loops" into "productive (but slower) data processing."
> >
> > **Table: Ablation study of SupervisorAgent's components regarding Latency** on the GAIA validation set.
> >
> > | Method                       | Avg. Latency (s)          |
> > |------------------------------|---------------------------|
> > | Smolagent                    | 233.96                    |
> > | + SMAS (w/o Correction)      | 280.77 ↑20.01%            |
> > | + SMAS (w/o Guidance)        | 271.95 ↑16.24%            |
> > | + SMAS (w/o Purification)    | 236.21 ↑0.96%             |
> > | + SMAS                       | 321.15 ↑37.27%            |

---

> > > ### Author Response · Authors · 2025-11-23
> > > **Response to Reviewer vopL (3/n)**
> > >
> > > ### **To Q2: Variant for Robust Baselines**.
> > >
> > > > The ablation study (Table 3) shows that "Purification" is the main source of token savings, while "Correction" and "Guidance" are key for accuracy . This suggests that for tasks where the baseline is already robust (high accuracy), a "Purification-Only" supervisor might offer the best cost-benefit. Have you considered this variant?
> > >
> > > This is an insightful suggestion. We argue that our **Adaptive Filter inherently realizes this "Purification-Only" variant dynamically**. Since the filter is heuristic-based (as detailed in Section 4.2), if a robust baseline (like AWorld) makes fewer errors or inefficient loops, the Correction and Guidance modules are simply **not triggered**. The Supervisor naturally shifts its focus almost exclusively to Purification, incurring zero overhead for the unused modules.
> > >
> > > **Empirical Evidence on Robust Baselines**. We validate this behavior by integrating the full SupervisorAgent into **OAgents** and **AWorld**, which are significantly more robust than the baseline Smolagent.
> > >
> > > - **Results**. As shown in Table 4 (line 465), **OAgents** achieves the exact same accuracy (46.67%) with and without SMAS, but SMAS reduces token costs by **39.36%**.
> > >
> > > - **Conclusion**. This confirms that for robust agents, the Supervisor automatically acts as an efficiency optimizer (Purification) without disrupting the base agent's strong reasoning capabilities, proving that manual removal of modules is unnecessary due to the system's adaptive nature.
> > >
> > > **Table: Cross-framework performance of SupervisorAgent on GAIA subset** (top-10 most token-intensive tasks per level).
> > >
> > > | Method                       | Avg. Acc.     | Avg. Token   | L1 Avg. Token | L2 Avg. Token | L3 Avg. Token |
> > > |------------------------------|---------------|--------------|--------------------|--------------------|--------------------|
> > > | Smolagent                    | 40.00         | 1,446,526    | 933,013            | 2,037,437          | 1,369,131          |
> > > | **+ SMAS**                   | 46.67 ↑6.67%  | 721,332 ↓50.13% | 522,364 ↓44.01%   | 960,694 ↓52.85%    | 680,939 ↓50.26%    |
> > > | AWorld (without Guard)       | 23.33         | 155,239      | 50,851             | 217,332            | 166,500            |
> > > | AWorld (with Guard)          | 30.00         | 353,738      | 135,413            | 463,083            | 376,878            |
> > > | **AWorld (with SMAS)**       | 36.67 ↑6.67%  | 224,480 ↓36.54% | 90,569 ↓33.12%    | 355,051 ↓23.33%    | 194,561 ↓48.38%    |
> > > | OAgents                      | 46.67         | 530,939      | 430,852            | 359,511            | 802,454            |
> > > | **+ SMAS**                   | 46.67         | 321,957 ↓39.36% | 214,604 ↓50.19%   | 274,875 ↓23.54%    | 476,393 ↓40.63%    |
> > >
> > > ### **To Q3: Sensitivity to Observation Threshold**.
> > >
> > > > The "Excessive Observation Length" trigger is a hardcoded "3,000 characters". How sensitive is the framework's performance (both token savings and accuracy) to this hyperparameter?
> > >
> > > We have addressed the concern regarding the "3,000 characters" threshold ($\tau_{\text{len}}$) via a dedicated sensitivity analysis in **Appendix A.3.2** (line 1024) and **Figure 5**.
> > >
> > > - **Stability**. Testing $\tau_{\text{len}}$ from **1,000 to 7,000** reveals that the framework is **highly robust**, with token savings and accuracy remaining stable across this range.
> > >
> > > - **Selection Logic**. Although $\tau_{\text{len}}=1000$ showed a slight peak, we selected **3,000** as the default to conservatively balance sensitivity (filtering noise) against specificity (preserving context).
> > >
> > > **Adaptability to other MAS**. Furthermore, we demonstrate that this threshold is adaptable rather than rigid. For the **OAgents** framework, which inherently generates more verbose outputs (complex tool usage), we adjusted $\tau_{\text{len}}$ to **10,000**. This successfully triggered the same efficiency gains, confirming that the parameter is logically correlated with the MAS's "verbosity" and can be easily tuned for different architectures.

---

> > > > ### Comment · Reviewer_vopL · 2025-11-25
> > > >
> > > > Thank you for your rebuttal. I think my rating is reasonable. I increased my confidence score.

---

### Official Review · Reviewer_hEWr · 2025-11-01

**Soundness:** 4
**Presentation:** 3
**Contribution:** 3
**Rating:** 6
**Confidence:** 4

**Summary:**

The paper addresses two critical challenges in MASs, 1) Operational Inefficiency (measured through token usage here), 2) Robustness Failures (error propagation, misinformation, etc.). Unlike prior methods that focus on post-hoc error attribution and diagnosis, the paper introduces SupervisorAgent, a meta-agent, that monitors the interactions in an MAS (agent-agent, agent-tool, agent-memory), uses an LLM-free "adaptive-filter" to determine high-risk interactions, and if invoked, the SupervisorAgent intervenes through actions like proactively correcting errors, guiding agents away from inefficient behaviors (like loops), and purifying excessively long or noisy observations to reduce token load.

The authors implement the SupervisorAgent over the SmolAgents framework, and compare against strong baselines across several benchmarks, finding that the intervention reduces token consumption by average as much as 29.45% without losing accuracy. The paper concludes with an ablation study of the various SupervisorAgent interventions, finding that purification leads to efficiency gains, whereas correction and guidance are necessary for performance.

**Strengths:**

1. The paper introduces a novel, lightweight, non-intrusive monitoring of MAS. This could potentially become a major MAS development pattern going ahead.
2. The paper is well-to-read, and educational from the perspective of understanding MAS design, issues with current MAS, and nicely introduces the proposed intervention.
3. Evaluation: The authors validate the intervention across 6 benchmarks, and across 3 leading LLMs, including open and proprietary models.

**Weaknesses:**

1. The overhead introduced due to SupervisorAgent is not described in detail.
2. The details about the working of adaptive filter are not described. Since one of the core features of the SupervisorAgent is "lightweight" monitoring, the authors should clearly describe how the adaptive filter works without LLMs, especially to identify "inefficient behavior" which seems to be a highly subjective criteria unlike the other 2 high-risk interactions identified.
3. Many MAS proposed in the past have included an additional agent that acts as a verifier or quality-assurance agent (for example, ChatDev includes a "Tester" agent). However, one issue with such interventions is that often the verifier can itself fail, introduce misinformation, or propagate errors. The authors do not discuss the failure modes or failure rate of complex SupervisorAgent actions like "correct_observation".

**Questions:**

1. Do the reported token costs for SMAS in all figures/tables take into account all supervisor token costs?
2. Can the authors describe the overhead introduced by SupervisorAgent, not only in terms of the token costs, but also in terms of wall-clock times, or number of additional LLM calls invoked (especially relevant since once a SupervisorAgent intervention will be invoked, the agent system's execution will need to be halted).
3. Can the authors' comment on the interaction of SupervisorAgent when implemented in other MAS frameworks, which could include significant interaction with external environments (for example, through MCP)?

---

> ### Author Response · Authors · 2025-11-23
> **Response to Reviewer hEWr (1/n)**
>
> We sincerely thank the reviewer for the positive assessment and for identifying our work as a potential "major MAS development pattern". The reviewer's insightful comments have guided us to significantly strengthen our evaluation. In the revised PDF, major updates are marked in **blue**.
>
> ### **To W1 & Q1: Token Overhead and Net Savings**.
>
> > The overhead introduced due to SupervisorAgent is not described in detail.
>
> > Do the reported token costs for SMAS in all figures/tables take into account all supervisor token costs?
>
> All reported token costs in the main results are "**Net Costs**," accounting for the Supervisor's overhead. To make this explicit, we have added a detailed breakdown of the Token Reduction Rate (TRR) versus **Net TRR** in **Table 6** and **Figure 6** (detailed in lines 480 & 1133). As shown in the table below, even after deducting the Supervisor's consumption, our method achieves significant net savings (e.g., 29.68% on GAIA pass@1).
>
> **Table: Token efficiency analysis on GAIA validation set.** Comparison of token consumption across different pass@k settings.
>
> | Method                              | Avg. Tokens (K)            | L1 Tokens (K)              | L2 Tokens (K)              | L3 Tokens (K)              |
> |-------------------------------------|----------------------------|----------------------------|----------------------------|----------------------------|
> | **pass@1**                          |                            |                            |                            |                            |
> | Smolagent                           | 527.76                     | 298.51                     | 619.59                     | 691.33                     |
> | + SMAS                              | 314.07 ↓40.49%             | 220.63 ↓26.09%             | 342.18 ↓44.77%             | 411.58 ↓40.47%             |
> | + SMAS (NET)                        | 371.12 ↓29.68%             | 258.28 ↓13.48%             | 404.96 ↓34.64%             | 489.22 ↓29.23%             |
> | **pass@2**                          |                            |                            |                            |                            |
> | Smolagent                           | 467.19                     | 275.85                     | 548.02                     | 589.92                     |
> | + SMAS                              | 329.51 ↓29.47%             | 231.96 ↓15.91%             | 354.21 ↓35.37%             | 446.64 ↓24.29%             |
> | + SMAS (NET)                        | 389.55 ↓16.62%             | 270.07 ↓2.10%              | 420.97 ↓23.18%             | 529.20 ↓10.29%             |
> | **pass@3**                          |                            |                            |                            |                            |
> | Smolagent                           | 502.40                     | 282.14                     | 605.05                     | 611.87                     |
> | + SMAS                              | 312.06 ↓37.89%             | 236.28 ↓16.25%             | 342.36 ↓43.42%             | 366.31 ↓40.13%             |
> | + SMAS (NET)                        | 369.52 ↓26.45%             | 276.84 ↓1.88%              | 409.05 ↓32.39%             | 427.72 ↓30.10%             |
>
> ### **To W2: Adaptive Filter Details and Subjectivity**.
>
> > The details about the working of adaptive filter are not described. Since one of the core features of the SupervisorAgent is "lightweight" monitoring, the authors should clearly describe how the adaptive filter works without LLMs, especially to identify "inefficient behavior" which seems to be a highly subjective criteria unlike the other 2 high-risk interactions identified.
>
> We addressed the concern regarding the potential subjectivity of hyperparameter selection in **Appendix A.3.2**. We explicitly define the three critical hyperparameters governing the adaptive filter: $\tau_{\text{step}}$ and $\tau_{\text{loop}}$ modulate the detection of inefficient behaviors, while $\tau_{\text{len}}$ defines the threshold for identifying excessive observations. **Algorithm 1** details their implementation, and **Table 5** lists the specific settings used across diverse benchmarks.
>
> To demonstrate that our method is not overly sensitive to specific manual tuning, we conducted a **sensitivity analysis** (detailed in line 1024) on the observation threshold ($\tau_{\text{len}}$) from **1000 to 7000**. The results confirm the system's robustness across a wide range of values, mitigating concerns about parameter subjectivity.

---

> > ### Author Response · Authors · 2025-11-23
> > **Response to Reviewer hEWr (2/n)**
> >
> > ### **To W3: Failure Modes and Verifier Reliability**.
> >
> > > Many MAS proposed in the past have included an additional agent that acts as a verifier or quality-assurance agent (for example, ChatDev includes a "Tester" agent). However, one issue with such interventions is that often the verifier can itself fail, introduce misinformation, or propagate errors. The authors do not discuss the failure modes or failure rate of complex SupervisorAgent actions like "correct observation".
> >
> > Unlike ChatDev's reactive 'Tester' which verifies aggregated codebases, our SupervisorAgent monitors atomic interactions in real-time. This narrower scope significantly reduces supervision complexity and the risk of cascading errors compared to full-code verification, as evidenced by our maintained success rates.
> >
> > To address the reviewer's concern regarding specific failure modes, we have conducted a detailed analysis in **Appendix A.5**, categorized into three levels:
> >
> > 1. **Hallucination in Extreme Contexts**. When observations are extremely excessive (e.g., >200k), the Purification module may omit critical details or hallucinate during compression. We accept this trade-off because preventing context overflow (OOM) is more critical for task completion than preserving perfect granularity.
> >
> > 2. **Ineffective Guidance in Stubborn Loops**. For particularly non-responsive agents, the Supervisor may fail to break the deadlock or provide suboptimal advice, risking an infinite correction loop. To prevent this, we enforce a hard limit of two interventions per sub-task to ensure bounded latency and costs.
> >
> > 3. **Variance in Operational Efficiency**. While model-agnostic, operational efficiency varies across LLMs. Weaker models incur higher costs via frequent basic corrections, whereas stronger ones might bypass filters for subtle flaws. Thus, efficiency gains, unlike functionality, correlate with the backbone's capability.
> >
> > ### **To Q2: Latency and Wall-Clock Time**.
> >
> > > Can the authors describe the overhead introduced by SupervisorAgent, not only in terms of the token costs, but also in terms of wall-clock times, or number of additional LLM calls invoked (especially relevant since once a SupervisorAgent intervention will be invoked, the agent system's execution will need to be halted).
> >
> > We have included a **latency analysis** on the full GAIA validation set in **Table 7** and **Figure 7** (detailed in lines 483 and 1175). We report that integrating the supervisor introduces an average latency increase of 37.27% (absolute delay **< 1.5 mins per task**). Our ablation study (Table 7) reveals that the **Purification** module is the **primary driver** of this latency, indicating the overhead is strictly tied to the processing of excessive information.
> >
> > Regarding additional calls, our **Adaptive Filter** ensures the Supervisor only halts execution when necessary. It does not intervene in every step. On average, the Supervisor accounts for only **15.45%** of the total token consumption, indicating a proportional and bounded number of additional LLM invocations. We consider this temporal cost a justifiable trade-off for the significant economic savings and enhanced robustness.
> >
> > **Table: Ablation study of SupervisorAgent's components regarding Latency** on the full GAIA validation set.
> >
> > | Method                       | Avg. Latency (s)          |
> > |------------------------------|---------------------------|
> > | Smolagent                    | 233.96                    |
> > | + SMAS (w/o Correction)      | 280.77 ↑20.01%            |
> > | + SMAS (w/o Guidance)        | 271.95 ↑16.24%            |
> > | + SMAS (w/o Purification)    | 236.21 ↑0.96%             |
> > | + SMAS                       | 321.15 ↑37.27%            |

---

> > > ### Author Response · Authors · 2025-11-23
> > > **Response to Reviewer hEWr (3/n)**
> > >
> > > ### **To Q3: Generalization to Other MAS Frameworks**.
> > >
> > > > Can the authors' comment on the interaction of SupervisorAgent when implemented in other MAS frameworks, which could include significant interaction with external environments (for example, through MCP)?
> > >
> > > We demonstrate generalization by integrating SupervisorAgent into **AWorld** and **OAgents**, encompassing both tool-intensive and hierarchical architectures.
> > >
> > > - **MCP Implementation**. Specifically for **AWorld**, we implemented SupervisorAgent as an **MCP service** that broadcasts capabilities (e.g., Error Diagnosis, Workflow Efficiency). This allows the agent to dynamically route requests based on its execution state, rather than relying solely on heuristic triggers (detailed in **Appendix A.3.2**, line 1085).
> > >
> > > - **Performance**. As shown in **Table 4** (line 465), our method remains highly effective in these complex environments. It reduces token costs by **36.54%** (AWorld) and **39.36%** (OAgents) while maintaining or improving accuracy.
> > >
> > > - **Differentiation**. We have also clarified the distinction between our method and AWorld's internal "Guard Agent" in the **Discussion** (line 507), highlighting our focus on **system-wide efficiency** versus their focus on factual verification.
> > >
> > > **Table: Cross-framework performance of SupervisorAgent on GAIA subset** (top-10 most token-intensive tasks per level).
> > >
> > > | Method                       | Avg. Acc.     | Avg. Token   | L1 Avg. Token | L2 Avg. Token | L3 Avg. Token |
> > > |------------------------------|---------------|--------------|--------------------|--------------------|--------------------|
> > > | Smolagent                    | 40.00         | 1,446,526    | 933,013            | 2,037,437          | 1,369,131          |
> > > | **+ SMAS**                   | 46.67 ↑6.67%  | 721,332 ↓50.13% | 522,364 ↓44.01%   | 960,694 ↓52.85%    | 680,939 ↓50.26%    |
> > > | AWorld (without Guard)       | 23.33         | 155,239      | 50,851             | 217,332            | 166,500            |
> > > | AWorld (with Guard)          | 30.00         | 353,738      | 135,413            | 463,083            | 376,878            |
> > > | **AWorld (with SMAS)**       | 36.67 ↑6.67%  | 224,480 ↓36.54% | 90,569 ↓33.12%    | 355,051 ↓23.33%    | 194,561 ↓48.38%    |
> > > | OAgents                      | 46.67         | 530,939      | 430,852            | 359,511            | 802,454            |
> > > | **+ SMAS**                   | 46.67         | 321,957 ↓39.36% | 214,604 ↓50.19%   | 274,875 ↓23.54%    | 476,393 ↓40.63%    |

---

### Author Response · Authors · 2025-12-03
**Rebuttal Summary for the Area Chair**

Dear Area Chair,

We sincerely thank you for managing the review process under these challenging circumstances. To assist in your final assessment, we provide a summary of the consensus that was forming during the rebuttal period and the significant improvements made to the manuscript.

**Positive Momentum and Consensus.** Our initial ratings were **6/6/6/4**. Notably, during the rebuttal phase (before the system reversion), **two positive reviewers (Scores: 6) explicitly raised their confidence scores to 4**, signaling that our responses and additional experiments had successfully solidified their assessment of our work's contribution.

**Summary of Key Rebuttal Actions.** We have diligently addressed all concerns raised by the reviewers. Major updates include:

- **Expanded Generalization to SOTA Baselines:** We embraced the challenge of testing tool-heavy SOTA frameworks. We integrated SupervisorAgent into **AWorld** and **OAgents**. Results (New **Table 4**) confirm our method reduces token costs by **36.54%** and **39.36%**, respectively, on these robust baselines, proving our approach is a universal efficiency layer, not merely a patch for weak MASs.

- **Transparency in Net Savings & Latency:** We provided a strict accounting of "**Net Token Savings**" in **Table 6**, confirming a net reduction of \~30%. Regarding **latency**, we clarified that the intrinsic monitoring overhead is **negligible** (+0.96%). The observed total increase (\~37%) stems almost exclusively from the compute-intensive *Purification* module processing massive contexts, which is a justifiable trade-off for the substantial token savings.

- **Rigorous Robustness & Consistency Checks:**

  - We replaced the subset ablation with a **Full GAIA Validation Set** study (New Table 3), confirming that *Purification* remains the consistent driver of efficiency across the general task distribution.

  - We conducted a **sensitivity analysis** on hyperparameters (Figure 5) and a detailed **Failure Mode Analysis** (Appendix A.5), demonstrating the system's adaptability.

- **Clarification of Conceptual Novelty:** We sharpened our positioning against prior "routers," defining **SupervisorAgent** as the pioneering efficiency-centric, conditional interventionist. We formalized this as the **Supervised MAS (SMAS)** paradigm—a foundational component for future scalable systems.

**Conclusion.** We believe the rebuttal has adequately resolved concerns regarding **generalization**, **conceptual novelty**, and **methodological robustness** (including parameter sensitivity and performance consistency). Given the strong initial support and the increased confidence from the positive reviewers, we believe the revised manuscript merits acceptance to ICLR 2026.

Thanks again for your time and dedication.

Sincerely,

The Authors

---

### Meta-Review · Area_Chair_CmyT · 2026-01-06

**Summary:**

The paper proposes SupervisorAgent, a lightweight, modular framework for runtime supervision in multi-agent systems (MAS), aiming to reduce token consumption and improve robustness through an LLM-free adaptive filter that triggers targeted interventions. All four reviewers acknowledged the relevance of the problem and the practicality of the design, particularly the hybrid “LLM-free filter + LLM supervisor” architecture. Initial scores were 6/6/6/4, reflecting general support but with reservations.

Key concerns centered on: (1) whether reported token savings accounted for the supervisor’s overhead; (2) latency and wall-clock time implications; (3) generalizability beyond the Smolagent baseline to stronger, tool-heavy MAS like AWorld or OAgents; (4) robustness of heuristic thresholds in the adaptive filter; and (5) conceptual novelty relative to prior meta-agent or routing approaches.

During rebuttal, the authors provided substantial new evidence: net token savings (including supervisor cost), latency breakdowns, full-dataset ablation studies, sensitivity analyses of hyperparameters, and successful integration into AWorld and OAgents—demonstrating consistent efficiency gains (~36–39% token reduction) without accuracy loss. These updates directly addressed the core methodological and empirical concerns raised by all reviewers.

**Reviewer Concerns:**

Reviewer hEWr (Rating: 6):
Concerns about token overhead, latency, failure modes, and generalization were thoroughly addressed. The authors provided net token accounting, latency ablation showing purification as the main source of delay, failure mode analysis, and cross-framework validation. These responses appear sufficient to resolve the reviewer’s doubts.

Reviewer vopL (Rating: 6):
Primary concerns were net token savings and latency. The authors confirmed all reported savings are net-inclusive and showed that architectural overhead (excluding purification) adds only ~0.96% latency. The reviewer explicitly increased their confidence post-rebuttal, indicating concerns were resolved.

Reviewer UCU6 (Rating: 6):
Raised critical questions about Pareto claims vs. SOTA (e.g., AWorld) and lack of SOTA integration. The authors responded with new experiments integrating SupervisorAgent into AWorld and OAgents, demonstrating further token reduction while maintaining or improving accuracy. They also replaced the subset ablation with a full GAIA validation study and provided sensitivity analysis—effectively addressing robustness and generalizability concerns.

Reviewer WiAV (Rating: 4):
Expressed skepticism about conceptual novelty, applicability to tool-heavy systems, and heuristic brittleness. The authors clarified the efficiency-centric runtime control paradigm (distinct from accuracy-focused supervisors), validated performance on tool-intensive frameworks (OAgents, AWorld), and provided both sensitivity analysis and failure mode characterization across model backbones. While conceptual positioning remains incremental to some degree, the empirical rigor and breadth of validation significantly strengthen the contribution.

All major technical and empirical concerns were substantively addressed. Remaining reservations (e.g., incremental novelty) do not outweigh the demonstrated utility and robustness of the approach.

**Reviewer Scores:**

Reviewer hEWr: Would likely maintain or slightly increase score to 6 (already confident; rebuttal solidified assessment).

Reviewer vopL: Explicitly increased confidence; would likely keep score at 6.

Reviewer UCU6: Concerns about SOTA comparison were central to their evaluation; new results directly counter the main weakness. Likely would raise score to 6 or possibly 8, but conservatively estimated as 6 given initial caution.

Reviewer WiAV: Initial score was 4 due to unresolved generalization and novelty concerns. Rebuttal provided strong empirical evidence on tool-heavy systems and clarified conceptual framing. Likely would revise score upward to 6.

---

### Decision · Program_Chairs · 2026-01-26

Accept (Poster)